# Understanding Influence Functions and Data-models via Harmonic Analysis

**Nikunj Saunshi, Arushi Gupta, Mark Braverman, Sanjeev Arora**
Department of Computer Science, Princeton University
{nsaunshi, arushig, mbraverm, arora}@cs.princeton.edu

## Abstract

Influence functions estimate effect of individual data points on predictions of the model on test data and were adapted to deep learning in Koh & Liang (2017). They have been used for detecting data poisoning, detecting helpful and harmful examples, influence of groups of datapoints, etc. Recently, Ilyas et al. (2022) introduced a linear regression method they termed *datamodels* to predict the effect of training points on outputs on test data. The current paper seeks to provide a better theoretical understanding of such interesting empirical phenomena. The primary tool is harmonic analysis and the idea of *noise stability*. Contributions include: (a) Exact characterization of the learnt datamodel in terms of Fourier coefficients. (b) An efficient method to estimate the residual error and quality of the optimum linear datamodel without having to train the datamodel. (c) New insights into when influences of groups of datapoints may or may not add up linearly.

## 1 Introduction

It is often of great interest to quantify how the presence or absence of a particular training data point affects the trained model's performance on test data points. Influence functions is a classical idea for this (Jaeckel, 1972; Hampel, 1974; Cook, 1977) that has recently been adapted to modern deep models and large datasets Koh & Liang (2017). Influence functions have been applied to explain predictions and produce confidence intervals (Schulam & Saria, 2019), investigate model bias (Brunet et al., 2019; Wang et al., 2019), estimate Shapley values (Jia et al., 2019; Ghorbani & Zou, 2019), improve human trust (Zhou et al., 2019), and craft data poisoning attacks (Koh et al., 2019).

Influence actually has different formalizations. The classic calculus-based estimate (henceforth referred to as *continuous influence*) involves conceptualizing training loss as a weighted sum over training datapoints, where the weighting of a particular datapoint $z$ can be varied infinitesimally. Using gradient and Hessian one obtains an expression for the rate of change in test error (or other functions) of $z'$ with respect to (infinitesimal) changes to weighting of $z$. Though the estimate is derived only for infinitesimal change to the weighting of $z$ in the training set, in practice it has been employed also as a reasonable estimate for the *discrete* notion of influence, which is the effect of completely adding/removing the data point from the training dataset (Koh & Liang, 2017). Informally speaking, this discrete influence is defined as $f(S \cup \{i\}) - f(S)$ where $f$ is some function of the test points, $S$ is a training dataset and $i$ is the index of a training point. (This can be noisy, so several papers use *expected influence* of $i$ by taking the *expectation* over random choice of $S$ of a certain size; see Section 2.) Koh & Liang (2017) as well as subsequent papers have used continuous influence to estimate the effect of decidedly non-infinitesimal changes to the dataset, such as changing the training set by adding or deleting entire groups of datapoints (Koh et al., 2019). Recently Bae et al. (2022) show mathematical reasons why this is not well-founded, and give a clearer explanation (and alternative implementation) of Koh-Liang style estimators.

Yet another idea related to influence functions is *linear datamodels* in Ilyas et al. (2022). By training many models on subsets of $p$ fraction of datapoints in the training set, the authors show that some interesting measures of test error (defined using logit values) behave as follows: the measure $f(x)$ is well-approximable as a (sparse) linear expression $\theta_0 + \sum_i \theta_i x_i$, where $x$ is a binary vector denoting a sample of $p$ fraction of training datapoints, with $x_i = 1$ indicating presence of $i$-th training point and $x_i = -1$ denoting absence. The coefficients $\theta_i$ are estimated via lasso

regression. The surprise here is that $f(x)$ —which is the result of deep learning on dataset $x$—is well-approximated by $\theta_0 + \sum_i \theta_i x_i$. The authors note that the $\theta_i$'s can be viewed as heuristic estimates for the discrete influence of the $i$th datapoint.

The current paper seeks to provide better theoretical understanding of above-mentioned phenomena concerning discrete influence functions. At first sight this quest appears difficult. The calculus definition of influence functions (which as mentioned is also used in practice to estimate the discrete notions of influence) involves Hessians and gradients evaluated on the trained net, and thus one imagines that any explanation for properties of influence functions must await better mathematical understanding of datasets, net architectures, and training algorithms.

Surprisingly, we show that the explanation for many observed properties turns out to be fairly generic. Our chief technical tool is harmonic analysis, and especially theory of *noise stability* of functions (see O'Donnell (2014) for an excellent survey).

### 1.1 OUR CONCEPTUAL FRAMEWORK (DISCRETE INFLUENCE)

Training data points are numbered 1 through $N$, but the model is being trained on a random subset of data points, where each data point is included independently in the subset with probability $p$. (This is precisely the setting in linear datamodels.) For notational ease and consistency with harmonic analysis, we denote this subset by $x \in \{-1, +1\}^N$ where $+1$ means the corresponding data point was included. We are interested in some quantity $f(x)$ associated with the trained model on one or more test data points. Note $f$ is a probabilistic function of $x$ due to stochasticity in deep net training – SGD, dropout, data augmentation etc. – but one can average over the stochastic choices and think of $f$ as deterministic function $f \colon \{\pm 1\}^N \to \mathbb{R}$. (In practice, this means we estimate $f(x)$ by repeating the training on $x$, say, 10 to 50 times.)

This scenario is close to classical study of boolean functions via harmonic analysis, except our function is real-valued. Using those tools we provide the following new mathematical understanding:

1. We give reasons for existence of *datamodels* of Ilyas et al. (2022), the phenomenon that functions related to test error are well-approximable by a linear function $\theta_0 + \sum_i \theta_i x_i$. See Section 3.1.

2. Section 2 gives exact characterizations of the $\theta_i$'s for data models with/without regularization. (Earlier, Ilyas et al. (2022) noted this for a special case: $p = 0.5$, $\ell_2$ regularization)

3. Using our framework, we give a new algorithm to estimate the degree to which a test function $f$ is well-approximated by a linear datamodel, *without having to train the datamodel per se*. See Section 3.2, where our method needs only $\mathcal{O}(1/\epsilon^3)$ samples instead of $\mathcal{O}(N/\epsilon^2)$.

4. We study *group influence*, which quantifies the effect of adding or deleting a set $I$ of datapoints to $x$. Ilyas et al. (2022) note that this can often be well-approximated by linearly adding the individual influences of points in $I$. Section 4 clarifies simple settings where linearity would fail, by a factor exponentially large in $|I|$, and also discusses potential reasons for the observed linearity.

### 1.2 OTHER RELATED WORK

Narasimhan et al. (2015) investigate when influence is PAC learnable. Basu et al. (2020) use second order influence functions and find they make better predictions than first order influence functions. Cohen et al. (2020) use influence functions to detect adversarial examples. Kong et al. (2021) propose an influence based re-labeling function that can relabel harmful examples to improve generalization instead of just discarding them. Zhang & Zhang (2022) use *Neural Tangent Kernels* to understand influence functions rigorously for highly overparametrized nets.

Pruthi et al. (2020) give another notion of influence by tracing the effect of data points on the loss throughout gradient descent. Chen et al. (2020) define multi-stage influence functions to trace influence all the way back to pre-training to find which samples were most helpful during pre-training. Basu et al. (2021) find that influence functions are fragile, in the sense that the quality of influence estimates depend on the architecture and training procedure. Alaa & Van Der Schaar (2020) use higher order influence functions to characterize uncertainty in a jack-knife estimate. Teso et al. (2021) introduce Cincer, which uses influence functions to identify suspicious pairs of examples for interactive label cleaning. Rahaman et al. (2019) use harmonic analysis to decompose a neural network into a piecewise linear Fourier series, thus finding that neural networks exhibit spectral bias.

Other instance based interpretability techniques include Representer Point Selection (Yeh et al., 2018), Grad-Cos (Charpiat et al., 2019), Grad-dot (Hanawa et al., 2020), MMD-Critic (Kim et al., 2016), and unconditional counter-factual explanations (Wachter et al., 2017).

Variants on influence functions have also been proposed, including those using Fisher kernels (Khanna et al., 2019), tricks for faster and more scalable inference (Guo et al., 2021; Schioppa et al., 2022), and identifying relevant training samples with relative influence (Barshan et al., 2020). Discrete influence played a prominent role in the surprising discovery of *long tail phenomenon* in Feldman (2020); Feldman & Zhang (2020): the experimental finding that in large datasets like ImageNet, a significant fraction of training points are *atypical*, in the sense that the model does not easily learn to classify them correctly if the point is removed from the training set.

## 2 HARMONIC ANALYSIS, INFLUENCE FUNCTIONS AND DATAMODELS

In this section we introduce notations for the standard harmonic analysis for functions on the hypercube (O'Donnell, 2014), and establish connections between the corresponding fourier coefficients, discrete influence of data points and linear datamodels from Ilyas et al. (2022).

### 2.1 PRELIMINARIES: HARMONIC ANALYSIS

In the conceptual framework of Section 1.1, let $[N] := \{1, 2, 3, ..., N\}$. Viewing $f: \{\pm 1\}^N \to \mathbb{R}$ as a vector in $\mathbb{R}^{2^N}$, for any distribution $\mathcal{D}$ on $\{\pm 1\}^N$, the set of all such functions can be treated as a vector space with inner product defined as $\langle f, g \rangle_{\mathcal{D}} = \mathbb{E}_{x \sim \mathcal{D}}[f(x)g(x)]$, leading to a norm defined as $\|f\|_{\mathcal{D}} = \sqrt{\mathbb{E}_{x \sim \mathcal{D}}[f(x)^2]}$. Harmonic analysis involves identifying special orthonormal bases for this vector space. We are interested in $f$'s values at or near *p-biased* points $x \in \{\pm 1\}^N$, where $x$ is viewed as a random variable whose each coordinate is independently set to $+1$ with probability $p$. We denote this distribution as $\mathcal{B}_p$. Properties of $f$ in this setting are best studied using the *orthonormal basis* functions $\{\phi_S : S \subseteq [N]\}$ defined as $\phi_S(x) = \prod_{i \in S} \left( \frac{x_i - \mu}{\sigma} \right)$, where $\mu = 2p - 1$ and $\sigma^2 = 4p(1-p)$ are the mean and variance of each coordinate of $x$. Orthonormality implies that $\mathbb{E}_x[\phi_S(x)] = 0$ when $S \neq \emptyset$ and $\langle \phi_S, \phi_{S'} \rangle_{\mathcal{B}_p} = \mathbb{1}_{S=S'}$. Then every $f: [N] \to \mathbb{R}$ can be expressed as $f = \sum_{S \subseteq [N]} \widehat{f}_S \phi_S$. Our discussion will often refer to $\widehat{f}_S$'s as "Fourier" coefficients of $f$, when the orthonormal basis is clear from context. This also implies Parseval's identity: $\sum_S \widehat{f}_S^2 = \|f\|_{\mathcal{B}_p}^2$. For any vector $\boldsymbol{z} \in \mathbb{R}^d$, we denote $\boldsymbol{z}_i$ to denote its $i$-th coordinate (with 1-indexing). For a matrix $\boldsymbol{A} \in \mathbb{R}^{d \times r}$, $\boldsymbol{A}_{i,:} \in \mathbb{R}^r$ and $\boldsymbol{A}_{:,j} \in \mathbb{R}^d$ denote its $i$-th row and $j$-th column respectively. We use $\| \cdot \|$ to denote Euclidean norm when not specified.

### 2.2 INFLUENCE FUNCTIONS

We use the notion of influence of a single point from Feldman & Zhang (2020); Ilyas et al. (2022). Influence of the $i$-th coordinate on $f$ at $x$ is defined as $Inf_i(f(x)) = f(x|_{i \to 1}) - f(x|_{i \to -1})$, where $x$ is sampled as a $p$-biased training set and $x|_{i \to 1}$ is $x$ with the $i$-th coordinate set to 1.

**Proposition 2.1** (Individual influence). *The "leave-one-out" influences satisfy the following:*

$$Inf_i(f(x)) = \frac{2}{\sigma} \sum_{S \ni i} \widehat{f}_S \phi_{S \setminus \{i\}}(x), \quad Inf_i(f) := \mathbb{E}_x[Inf_i(f(x))] = \frac{2}{\sigma} \widehat{f}_{\{i\}}. \tag{1}$$

Thus the degree-1 Fourier coefficients are directly related to average influence of individual points. Similar results can be shown for other definition of single point influence: $\mathbb{E}_x[f(x) - f(x|_{i \to -1})]$ and $\mathbb{E}_x[f(x|_{i \to 1}) - f(x)]$ are equal to $p\, Inf_i(f)$ and $(1-p)\, Inf_i(f)$ respectively. The proof of this follows by observing that $\mathbb{E}_x[f(x|_{i \to 1}) - f(x|_{i \to -1})] = \sum_{S \ni i} \widehat{f}_S \left( \frac{1-\mu}{\sigma} - \frac{-1-\mu}{\sigma} \right) \phi_{S \setminus \{i\}}(x)$. The only term that is not zero in expectation is the one for $S = \{i\}$, thus proving the result. Section 4 deals with the influence of add or deleting larger subsets of points.

**Continuous vs Discrete Influence** Koh & Liang (2017) utilize a continuous notion of influence: train a model using dataset $x \in \{\pm 1\}^N$, and then treat the $i$-th coordinate of $x$ as a continuous variable $x_i$. Compute $\frac{d}{dx_i} f$ at $x_i = 1$ using gradients and Hessians of the loss at end of training.

This is called the *continuous influence* of the $i$-th datapoint on $f$. As mentioned in Section 1 in several other contexts one uses the discrete influence 1, which has a better connection to harmonic analysis. While experiments in Koh & Liang (2017) suggest that continuous influence closely tracks the discrete influence in linear models, Bae et al. (2022) show that this breaks in deep learning settings. For the rest of the paper we will use discrete influences.

## 2.3 LINEAR DATAMODELS FROM A HARMONIC ANALYSIS LENS

Next we turn to the phenomenon Ilyas et al. (2022) that a function $f(x)$ related to average test error[1] (where $x$ is the training set) often turns out to be approximable by a linear function $\theta_0 + \sum_i \theta_i \bar{x}_i$, where $\bar{x} \in \{0,1\}^N$ is the binary version of $x \in \{\pm 1\}^N$. It is important to note that this approximation (when it exists) holds only in a least squares sense, meaning that the following is small: $\mathbb{E}_x[(f(x) - \theta_0 - \sum_i \theta_i \bar{x}_i)^2]$ where the expectation is over $p$-biased $x$.

The authors suggest that $\theta_i$ can be seen as an estimate of the average discrete influence of variable $i$. While this is intuitive, they do not give a general proof (their Lemma 2 proves it for $p = 1/2$ with $\ell_2$ regularization). The following result exactly characterizes the solutions for arbitrary $p$ and with both $\ell_1$ and $\ell_2$ regularization.

**Theorem 2.2** (Characterizing solutions to linear datamodels). *Denote the quality of a linear datamodel $\theta \in \mathbb{R}^{N+1}$ on the $p$-biased distribution over training sets $\mathcal{B}_p$ by*

$$\mathcal{R}(\theta) := \mathbb{E}_{x \sim \mathcal{B}_p}\left[\left(f(x) - \theta_0 - \sum_{i=1}^N \theta_i \bar{x}_i\right)^2\right]. \tag{2}$$

*where $\bar{x} \in \{0,1\}^N$ is the binary version for any $x \in \{\pm 1\}^N$. Then for $\mu = 2p - 1$ and $\sigma = \sqrt{4p(1-p)}$, the following are true about the optimal datamodels with and without regularization:*

*(a) The unregularized minimizer $\theta^\star = \arg\min \mathcal{R}(\theta)$ satisfies*

$$\theta_i^\star = \frac{2}{\sigma}\widehat{f}_{\{i\}}, \quad \theta_0^\star = \widehat{f}_\emptyset - \frac{(\mu+1)}{2}\sum_i \theta_i^\star. \tag{3}$$

*Furthermore the residual error is the sum of all Fourier coefficients of order 2 or higher.*

$$\mathcal{R}(\theta^\star) = B_{\geq 2} := \sum_{S \subseteq [N]:|S|\geq 2} \widehat{f}_S^2 \tag{4}$$

*(b) The minimizer with $\ell_2$ regularization $\theta^\star(\lambda, \ell_2) = \arg\min\left\{\mathcal{R}(\theta) + \lambda\|\theta_{1:N}\|_2^2\right\}$ satisfies*

$$\theta^\star(\lambda, \ell_2)_i = \frac{2}{\sigma}\left(1 + \frac{4\lambda}{\sigma^2}\right)^{-1}\widehat{f}_{\{i\}}, \tag{5}$$

*(c) The minimizer with $\ell_1$ regularization $\theta^\star(\lambda, \ell_1) = \arg\min\left\{\mathcal{R}(\theta) + \lambda\|\theta_{1:N}\|_1\right\}$ satisfies*

$$\theta^\star(\lambda, \ell_1)_i = \frac{2}{\sigma}\left(\left(\widehat{f}_{\{i\}} - \lambda/\sigma\right)_+ - \left(-\widehat{f}_{\{i\}} - \lambda/\sigma\right)_+\right) = \frac{2}{\sigma}\,\text{sign}\left(\widehat{f}_{\{i\}}\right)\left(\left|\widehat{f}_{\{i\}}\right| - \lambda/\sigma\right)_+ \tag{6}$$

*where $z_+ = z\mathbb{1}_{z>0}$ is the standard ReLU operation.*

This result shows that the optimal linear datamodel with various regularization schemes, for any $p \in (0, 1)$ are directly related to the first order Fourier coefficients at $p$. Given that the average discrete influences, from Equation (1), are also the first order coefficients, this result directly establishes a connection between datamodels and influences. Result (c) suggests that $\ell_1$ regularization has the effect of clipping the Fourier coefficients such that those with small magnitude are set to 0, thus encouraging sparse datamodels. Furthermore, Equation (4) also gives a simple expression for the residual of the best linear fit which we utilize for our efficient residual estimation procedure in Section 3.2.

The proof of the full result is presented in Appendix B.1, however we present a proof sketch of the result (a) to highlight the role of the Fourier basis and coefficients.

---

[1]The average is not over test error but over the difference between the correct label logits and the top logit among the rest, at the output layer of the deep net.

*Proof sketch for Theorem 2.2(a).* Since $\{\phi_S\}_{S \subseteq [N]}$ is an orthonormal basis for the inner product space with $\langle f, g \rangle_{\mathcal{B}_p} = \mathbb{E}_{x \sim \mathcal{B}_p}[f(x)g(x)]$, we can rewrite $\mathcal{R}(\theta)$ as follows

$$\mathcal{R}(\theta) = \mathbb{E}_{x \sim \mathcal{B}_p}\left(f(x) - \theta_0 - \sum_i \theta_i \bar{x}_i\right)^2 = \mathbb{E}_{x \sim \mathcal{B}_p}\left(f(x) - \bar{\theta}_0 - \sum_i \bar{\theta}_i \phi_{\{i\}}(x)\right)^2$$

$$= \left\| f - \bar{\theta}_0 - \sum_i \bar{\theta}_i \phi_{\{i\}} \right\|_{\mathcal{B}_p}^2, \text{ where } \| \cdot \|_{\mathcal{B}_p} := \langle \cdot, \cdot \rangle_{\mathcal{B}_p}$$

where $\bar{\theta}_i = \sigma/2 \, \theta_i$ and $\bar{\theta}_0 = \theta_0 + (\mu+1)/2 \sum_i \theta_i$. Due to the orthonormality of $\{\phi_S\}_{S \subseteq [N]}$, the minimizer $\bar{\theta}^\star$ will lead to a projection of $f$ onto the span of $\{\phi_S\}_{|S| \leq 1}$, which gives $\bar{\theta}_i = \widehat{f}_{\{i\}}$ and thus $\theta_i = 2/\sigma \widehat{f}_{\{i\}}$. Furthermore the residual is the norm of the projection of $f$ onto the orthogonal subspace $\text{span}\left(\{\phi_S\}_{|S| \geq 2}\right)$, which is precisely $\sum_{|S| \geq 2} \widehat{f}_S^2$. $\qquad\square$

## 3 NOISE STABILITY AND QUALITY OF LINEAR DATAMODELS

Theorem 2.2 characterizes that best linear datamodel from Ilyas et al. (2022) for any the test function. The unanswered question is: *Why does this turn out to be a good surrogate (i.e. with low residual error) for the actual function $f$?* A priori, one expects $f(x)$, the result of deep learning on $x$, to be a complicated function. In this section we use the idea of *noise stability* to provide an intuitive explanation. Furthermore we show how noise stability can be leveraged for efficient estimation of the quality of fit of linear datamodels, without having to learn the datamodel (which, as noted in Ilyas et al. (2022), requires training a large number of nets corresponding to random training sets $x$).

**Noise stability.** Suppose $x$ is $p$-biased as above. We define a $\rho$-correlated r.v. as follows:

**Definition 3.1** ($\rho$-correlated). *For $x \in \{\pm 1\}^N$, we say a r.v. $x'$ is $\rho$-correlated to $x$ if it is sampled as follows: If $x_i = 1$, then $x_i' = -x_i$ w.p. $(1-\rho)(1-p)$. If $x_i = -1$, $x_i' = -x_i$ w.p. $(1-\rho)p$.*

Note that $x_i' = 1$ w.p. $p(1-(1-\rho)(1-p)) + (1-p)(1-\rho)p = p$, so both $x$ and $x'$ represent training datsets of expected size $pN$ with expected intersection of $(p + \rho(1-p))N$. Define the *noise-stability of $f$ at noise rate $\rho$* as $h(\rho) = \mathbb{E}_{x,x'}[f(x)f(x')]$ where $x, x'$ are $\rho$-correlated. Noise-stability plays a role reminiscent of moment generating function in probability, since ortho-normality implies

$$h(\rho) = \mathbb{E}_{x,x'}[f(x)f(x')] = \sum_S \widehat{f}_S^2 \rho^{|S|} = \sum_{i=1}^d B_i \rho^i, \text{ where } B_i = \sum_{S:|S|=i} \widehat{f}_S^2. \qquad (7)$$

Thus $h(\rho)$ is a polynomial in $\rho$ where the coefficient of $\rho^i$ captures the $\ell_2$ weight of coefficients corresponding to sets $S$ of size $i$.

### 3.1 WHY SHOULD LINEAR DATAMODELS BE A GOOD APPROXIMATION?

Suppose $f = \sum_S \widehat{f}_S \phi_S$ is the function of interest. Let $B_i$ stand for $\sum_{S:|S|=i} \widehat{f}_S^2$. Define the *normalized noise stability* as $\bar{h}(\rho) = \mathbb{E}_{x,x'}[f(x)f(x')] \, / \, \mathbb{E}_x[f(x)^2]$ for $\rho \in [0, 1]$. Intuitively, since $f$ concerns test loss, one expects that as the number of training samples grows, test behavior of two correlated data sets $x, x'$ would not be too different, since we can alternatively think of picking $x, x'$ as first randomly picking their intersection and then augmenting this common core with two disjoint random datasets. Thus intuitively, normalized noise stability should be high, and perhaps close to its maximum value of 1. If it is indeed close to 1 then the next theorem (a modification of a related theorem for boolean-valued functions in O'Donnell (2014)) gives a first-cut bound on the quality of the best linear approximation in terms of the magnitude of the residual error. (Note that linear approximation includes the case that $f$ is constant.)

**Theorem 3.1.** *The quality of the best linear approximation to $f$ can be bounded in terms of the normalized noise stability $\bar{h}$ as*

$$\text{(Normalized residual)} \quad \frac{\sum_{S:|S| \geq 2} \widehat{f}_S^2}{\sum_S \widehat{f}_S^2} \leq \frac{1 - \bar{h}(\rho)}{1 - \rho^2}. \qquad (8)$$

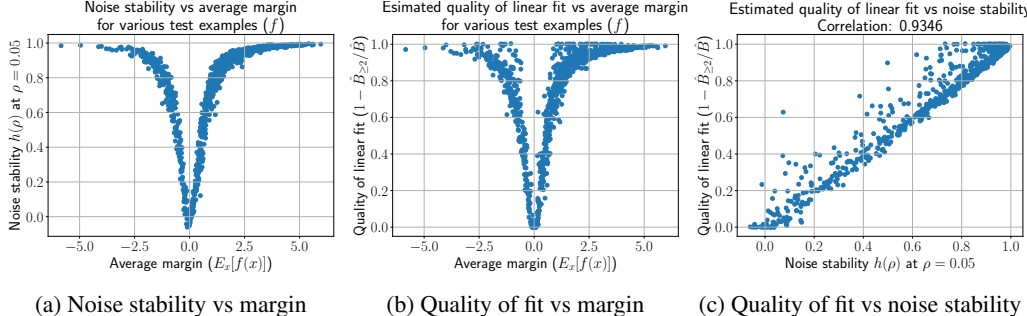

(a) Noise stability vs margin  (b) Quality of fit vs margin  (c) Quality of fit vs noise stability

Figure 1: Scatter plots for 1000 test examples and their margin functions $f$ (log of ratio of correct label probability and the best probability for another label), for networks trained on $p$-biased sets $x$. (a) The normalized noise stability from Theorem 3.1 is high for many points; $\approx 44\%$ points are higher than 0.9. Also, noise stability is low for points that have margin close to 0, i.e. those that are close to the decision boundary. (b) Similarly, the estimated quality of linear fit (using Algorithm 1) is good when the model is confidently correct or wrong. (c) Noise stability for $\rho = 0.05$ correlates highly with the estimated quality of linear fit, providing credibility to Theorem 3.1.

Theorem 3.1 is well-known in Harmonic analysis and is in some sense the best possible estimate if all we have is the noise stability for a single (and small) value of $\rho$. In fact we find in Figure 1c that the noise stability estimate does correlate strongly with the estimated quality of linear fit (based on our procedure from Section 3.2). Figure 3 plots noise stability estimates for small values of $\rho$.

In standard machine learning settings, it is not any harder to estimate $h(\rho)$ for more than one $\rho$, and this is is used in next section in a method to better estimate of the quality of linear approximation.

## 3.2 BETTER ESTIMATE OF QUALITY OF LINEAR APPROXIMATION

A simple way to test the quality of the best linear fit, as in Ilyas et al. (2022), is to learn a datamodel using samples and evaluate it on held out samples. However learning a linear datamodel requires solving a linear regression on $N$-dimensional inputs $x$, which could require $\mathcal{O}(N)$ samples in general. A sample here corresponds to training a neural net on a $p$-biased dataset $x$, and training $\mathcal{O}(N)$ such models can be expensive. (Ilyas et al. (2022) needed to train around a million models.)

Instead, can we estimate the quality of the linear fit without having to learn the best linear datamodel? This question relates to the idea of property testing in boolean functions, and indeed our next result yields a better estimate by using noise stability at multiple points. The idea is to leverage Equation (7), where the Fourier coefficients of sets of various sizes show up in the noise stability function $h(\rho) = \sum_{i=0}^{N} B_i \rho^i$ as the non-negative coefficients of the polynomial in $\rho$. Since the residual of the best linear datamodel, from Theorem 2.2, is the total mass of Fourier coefficients of sizes at least 2, i.e. $B_{\geq 2} = \sum_{i=2}^{N} B_i$, we can hope to learn this by fitting a polynomial on estimates of $h(\rho)$ at multiple $\rho$'s. Algorithm 1 precisely leverages by first estimating the degree 0 and 1 coefficients ($B_0$ and $B_1$) using noise stability estimates at a few $\rho$'s, estimating $B = \sum_{i=0}^{N} B_i = h(1)$ using $\rho = 1$, and finally estimating the residual using the expression $B_{\geq 2} = B - B_0 - B_1$. The theorem below shows that this indeed leads to a good estimate of the residual with having to train way fewer (independent of $N$) models.

**Theorem 3.2.** *Let $\hat{B}_{\geq 2} = \text{RESIDUALESTIMATION}(f, n, [0, \rho, 2\rho], 2)$ be the estimated residual (see Algorithm 1) after fitting a degree 2 polynomial to noise stability estimates at $0, \rho, 2\rho$, using $n$ calls to $f$. If $n = \mathcal{O}(1/\epsilon^3)$ and $\rho = \sqrt{\epsilon}$, then with high probability we have that $|\hat{B}_{\geq 2} - B_{\geq 2}| \leq \epsilon$.*

The proof of this is presented in Appendix B.2. This improves upon prior results on residual estimation for linear thresholds from Matulef et al. (2010) that do a degree-1 approximation to $h(\rho)$ with $1/\epsilon^4$ samples, more than $1/\epsilon^3$ samples needed with our degree-2 approximation instead. In fact, we hypothesize using a degree $d > 2$ likely improves upon the dependence on $\epsilon$; we leave that for future work. This result provides us a way to estimate the quality of the best linear datamodel without

---

**Algorithm 1** Efficient algorithm for residual estimation

---

1: **procedure** RESIDUALESTIMATION($f$, $n$, $[\rho_1, \ldots, \rho_k]$, $d$)
2:    /* $f$: function, $n$: evaluation budget, $d$: degree of approximation */
3:    $\boldsymbol{y} \in \mathbb{R}^k$, $\boldsymbol{A} \in \mathbb{R}^{k \times (d+1)}$
4:    **for** $i \in [1, \ldots, k]$ **do**
5:       $\boldsymbol{y}_i \leftarrow$ NOISESTABILITY($f$, $n/(k+1)$, $\rho_i$)           ▷ Estimate $h(\rho_i)$
6:       $\boldsymbol{A}_{i,:} \leftarrow [1, \rho_i, \ldots, \rho_i^d]$
7:    **end for**
8:    Solve $\hat{\boldsymbol{z}} = \min_{\boldsymbol{z} \in \mathbb{R}^{d+1}} \|\boldsymbol{A}\boldsymbol{z} - \boldsymbol{y}\|_2^2$ s.t. $\boldsymbol{z} \geq \boldsymbol{0}$    ▷ Requires solving a convex program
9:    $\hat{B}_0, \hat{B}_1 \leftarrow \hat{z}_1, \hat{z}_2$
10:    $\hat{B} \leftarrow$ NOISESTABILITY($f$, $n/(k+1)$, $1$)    ▷ Estimate $h(1) = \sum_i B_i$; see Equation (7)
11:    $\hat{B}_{\geq 2} \leftarrow \hat{B} - \hat{B}_0 - \hat{B}_1$
12:    **return** $\hat{B}_{\geq 2}$            ▷ $B_{\geq 2}$ is the residual from Theorem 2.2
13: **end procedure**
14:
15: **procedure** NOISESTABILITY($f$, $n$, $\rho$)
16:    $h_\rho \leftarrow 0$
17:    **for** $j \in [1, \ldots, n/2]$ **do**
18:       $x \sim \mathcal{B}_p$
19:       $x' \sim$ RHOCORR($x$, $\rho$)         ▷ $x'$ is $\rho$-correlated to $x$; see Definition 3.1
20:       $h_\rho \leftarrow h_\rho + 2/n f(x) f(x')$         ▷ Two evaluations of $f$ per $x$ needed
21:    **end for**
22:    **return** $h_\rho$       ▷ Returns an unbiased estimate for $h(\rho)$; see Lemma B.1
23: **end procedure**

---

having to use $\mathcal{O}(N/\epsilon^2)$ samples (in the worse case) for linear regression[2]. The guarantee does not even depend on $N$, although it has a worse dependence of $\epsilon$ that can likely be improved upon.

**Experiments.** We run our residual estimation algorithm for 1000 test examples (see Appendix C for details) and Figure 2 summarizes our findings. The histogram of the estimated normalized residuals (8) in Figure 2a indicates that a good linear fit exists for majority of the points, echoing the findings from Ilyas et al. (2022). Figures 2b and 2c study the effects of choices like degree $d$ and $\rho$. Furthermore we find in Figure 1b an interesting connection between the predicted quality of linear fit ($1-$ normalized residual) and the average margin of the test point: linear fit is best when models trained on $p$-biased datasets are confidently right or wrong on average. The fit is much worse for examples that are closer to the decision boundary (smaller margin); exploration of this finding is an interesting future direction. Finally, Figures 3 and 4 provide visualizations of the learned polynomial fits as the degree $d$ and list of $\rho$'s are varied.

# 4 UNDERSTANDING GROUP INFLUENCE AND ABILITY FOR COUNTERFACTUAL REASONING

Both influence functions as well as linear datamodels display some ability to do counterfactual reasoning: to predict the effect of small changes to the training set on the model's behavior on test data. Specifically, they allow reasonable estimation of the difference between a model trained with $x$ and one trained with $x|_{I \to -1}$, the training set containing $x$ after deleting the points in $I$. Averaged over $x$, this can be thought of as *average group influence* of deleting $I$. We study this effect through the lens of Fourier coefficients in the upcoming section.

## 4.1 EXPRESSION FOR GROUP INFLUENCE

Let $I$ be a subset of variables and let $x \in \{-1, 1\}^N$ denote a random $p$-biased variable with distribution $\mathcal{B}_p$. Then the expectation of $f(x) - f(x|_{I \to -1})$ can be thought of as the average

---

[2]If the datamodel is sparse, lasso can learn it with $\mathcal{O}(S \log N/\epsilon^2)$ samples, where $S$ is the sparsity.

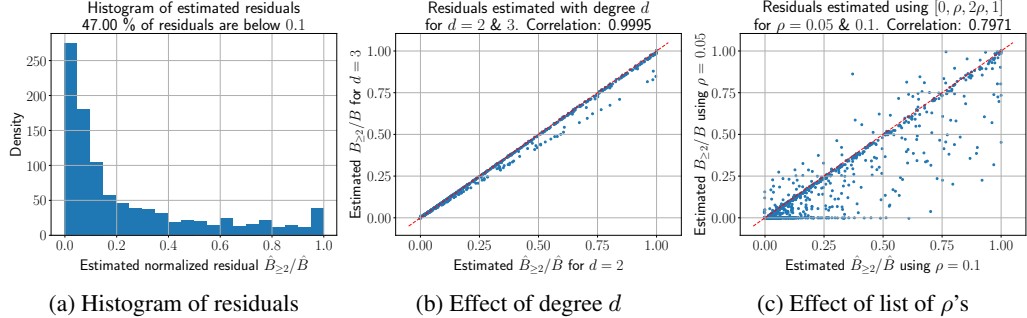

(a) Histogram of residuals  (b) Effect of degree $d$  (c) Effect of list of $\rho$'s

Figure 2: (a) Histogram of estimated normlized residual ($\hat{B}_{\geq 2}/\hat{B}$), for 1000 test examples, using Algorithm 1 with degree $d = 2$ and list $[0, 0.1, 0.2, 1]$ for $\rho$'s. Almost half of the test examples have normalized residuals below 0.1. (b) Estimations using $d = 2$ and $d = 3$, for various test points, are highly correlated to each other, suggesting a small effect of degree of approximation in this case. (c) Estimates using $[0, \rho, 2\rho, 1]$ in Algorithm 1, for $\rho = 0.05$ and $\rho = 0.1$; Spearman correlation is $\approx 0.8$. The choice of $\rho$ does have some effect, suggesting that there is still some noise in the estimate. Note that Theorem 3.2 requires the right scale of $\rho$ for the estimate to be accuracy.

*influence of deleting $I$.* An interesting empirical phenomenon in Koh et al. (2019) is that the group influence is often well-correlated with the sum of individual influences in $I$, i.e. $\sum_i \mathrm{Inf}_i(f(x))$. In fact this is what makes empirical study of group influence feasible in various settings, because influential subsets $I$ can be found without having to spend time exponential in $|I|$ to search among all subsets of this size. The claim in Ilyas et al. (2022) is that linear data models exhibit the same phenomenon: the sum of coefficients of coordinates in $I$ approximates the effect of training on $x|_{I \to -1}$. We give the mathematics of such "counterfactual reasoning" as well as its potential limits.

**Theorem 4.1.** *[Group influence] The following are true about group influence of deletion.*

$$\mathrm{Inf}_I(f) := \mathbb{E}_{x \sim \mathcal{B}_p}\left[f(x) - f(x|_{I \to -1})\right] = p \sum_{i \in I} \theta^\star - \sum_{I' \subseteq I, |I'| \geq 2} (-1)^{|I'|} \left(\frac{p}{1-p}\right)^{|I'|/2} \widehat{f}_{I'}$$

$$= p \sum_{i \in I} \mathrm{Inf}_i(f) - \sum_{I' \subseteq I, |I'| \geq 2} (-1)^{|I'|} \left(\frac{p}{1-p}\right)^{|I'|/2} \widehat{f}_{I'}$$

*where $\theta^\star$ is the optimal datamodel (Equation (3)) and $\mathrm{Inf}_i$ is an individual influence (Equation (1)).*

Thus we see the that average group influence of deletion[3] is equal to the sum of individual influence, plus a residual term. Proof for the above theorem for this is presented in Appendix B.3.

This residual term can in turn be upper bounded by $(1 - p)^{-|I|/2}\sqrt{B_{\geq 2}}$ (see Lemma B.2), which blows up exponentially in $|I|$. However the findings in Figure F.1 from Ilyas et al. (2022) suggest that the effect of number of points deleted is linear (or sub-linear), and far from an exponential growth. In the next section we provide a simple example where the exponential blow-up is unavoidable, and also provide some hypothesis and hints as to why this exponential blow is not observed in practice.

### 4.2 THRESHOLD FUNCTION AND EXPONENTIAL GROUP INFLUENCE

We exhibit the above exponential dependence on set size using a natural function inspired by empirics around the *long tail phenomenon* (Feldman, 2020; Feldman & Zhang, 2020), which suggests that successful classification on a specific test point depends on having seeing a certain number of "closely related" points during training. We model this as a sigmoidal function that depends on a special set $A$ of coordinates (i.e., training points) and assigns probability close to 1 when many more than $\beta$ fraction of the coordinates in $A$ are $+1$.

For any vector $z \in \{\pm 1\}^d$, let $\mathrm{avg}(z) = \frac{1}{d}\sum_{i=1}^d \frac{(1+z_i)}{2}$ denote the fraction of 1's in $z$ and $z_A = (z_i)_{i \in A}$ denotes the subset of coordinates indexed by $A \subseteq [d]$.

---

[3]Similar results can be shown for the average group influence of adding a set of points $I$.

**Example 1** (Sigmoid). *Consider a function $f(x) = S(\alpha(avg(x_A) - \beta))$ for a subset $A \subseteq [N]$ of size $M$, where $S(u) = (1 + e^{-u})^{-1}$ is the sigmoid function. The function $f$ could represent the probability of the correct label for a test point when trained on the training set $x \in \{\pm 1\}^N$.*

For this function, we show below that for a large enough $p$, the group influence of a $\gamma$-fraction subset of $A$ is exponential in the size of the set.

**Lemma 4.2.** *Consider the function[4] $f$ from Example 1 with $\beta = 0.5$ and $\alpha \to \infty$; so $f(x) = \mathbb{1}\{avg(x_A) > 0.5\}$. If $x$ is p-biased with $p > 0.5$, then for any constant fraction subset $A' \subseteq A$, its group influence is exponentially larger than sum of individual influences.*

$$Inf_{A'}(f) \geq \frac{(2(1-p))^{-|A'|+1}}{|A'|} \left( \sum_{i \in A'} Inf_i(f) \right). \tag{9}$$

Thus in the above example when $p > 0.5$, the group influence can be exponentially larger than what is captured by individual influences. Proof of this lemma is presented in Appendix B.3

**Margin v/s probability.** The function $f$ from Lemma 4.2 has a property that it saturates to 1 quickly, so once $p$ is large enough, the individual influence of deleting 1 point can be extremely small, but the group influence of deleting sufficiently many of the influential points can switch the probability to close to 0. Ilyas et al. (2022) however do not fit a datamodel to the probability and instead fit it to the "margin", which is the $\log$ of the ratio of probabilities of the correct label to the highest probability assigned to another label. Presumably this choice was dictated by better empirical results, and now we see that it may play an important role in the observed linearity of group influence. Specifically, their margin does not saturate and can get arbitrarily large as the probability of the correct label approaches 1. This can be seen by considering a slightly different (but related) notion of margin, defined as $\bar{f} = \log(f/(1-f))$, where the denominator is the *total probability* assigned to other labels instead of the max among other labels. For this case, the following result shows that group influence is now adequately represented by the individual influences.

**Corollary 4.3.** *For a function $f$ from Example 1, the group influence of any set $A' \subseteq [N]$ on the margin function $\bar{f}$ satisfies the following:*

$$Inf_{A'}(\bar{f}) = p \sum_{i \in A'} Inf_i(\bar{f}) \tag{10}$$

This result follows directly by observing that the margin function is simply the inverse of the sigmoid function, and so its expression is $\bar{f}(x) = \alpha(avg(x_A) - \beta)$ which is just a linear function. Since all the Fourier coefficients of sets of size 2 or larger will be zero for a linear function, the result follows from Theorem 4.1 where the residual term becomes 0.

## 5 CONCLUSION

This paper has shown how harmonic analysis can shed new light on interesting phenomena around influence functions. Our ideas use a fairly black box view of deep nets, which helps bypass the current incomplete mathematical understanding of deep learning. Our new algorithm of Section 3.2 has the seemingly paradoxical property of being able to estimate exactly the quality of fit of datamodels without actually having to train (at great expense) the datamodels. We hope this will motivate other algorithms in this space.

One limitation of harmonic analysis is that an arbitrary $f(x)$ could in effect behave very differently for on $p$-biased distributions at $p = 0.5$ versus $p = 0.6$. But in deep learning, the trained net probably does not change much upon increasing the training set by $20\%$. Mathematically capturing this stability over $p$ (not to be confused with noise stability of Equation (7) via a mix of theory and experiments promises to be very fruitful. It may also lead to a new and more fine-grained generalization theory that accounts for the empirically observed long-tail phenomenon. Finally, harmonic analysis is often the tool of choice for studying phase transition phenomena in random systems, and perhaps could prove useful for studying emergent phenomena in deep learning with increasing model sizes.

---

[4]The result can potentially be shown for more general $\alpha, \beta, p$

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

# A  ADDITIONAL BACKGROUND

## A.1  HARMONIC ANALYSIS AND BOOLEAN FUNCTIONS

We provide some more details for the classical Harmonic analysis on the boolean hypercube. Please refer to O'Donnell (2014) for an extensive discourse on this topic. The set of all functions $f\colon \{\pm 1\}^N \to \mathbb{R}$ can be trivially viewed as a vector space, since is it closed under linear combinations. Further for any distribution $\mathcal{D}$ over the hypercube $\{\pm 1\}^N$, the set of real valued functions can be viewed as an inner product space, where the inner product is defined as $\langle f, g \rangle_{\mathcal{D}} = \mathbb{E}_{x \sim \mathcal{D}}[f(x)g(x)]$. An alternative way to view this inner product is to define a $2^N$ dimensional vector $\boldsymbol{f} \in \mathbb{R}^{2^N}$ for every function $f$, where $\boldsymbol{f} = \left( \sqrt{\mathcal{D}(x)} f(x) \right)_{x \in \{\pm 1\}^N}$. Under this parameterization, the aforementioned inner product between $f$ and $g$ can be viewed as the standard dot product between $\boldsymbol{f}$ and $\boldsymbol{g}$, i.e. $\boldsymbol{f}^\top \boldsymbol{g} = \sum_{x \in \{\pm 1\}^N} \sqrt{\mathcal{D}(x)} f(x) \sqrt{\mathcal{D}(x)} g(x) = \sum_{x \in \{\pm 1\}^N} \mathcal{D}(x) f(x) g(x) = \mathbb{E}_{x \sim \mathcal{D}}[f(x)g(x)] = \langle f, g \rangle_{\mathcal{D}}$. This inner product leads to the norm defined as $\|f\|_{\mathcal{D}} = \sqrt{\mathbb{E}_{x \sim \mathcal{D}}[f(x)^2]}$.

Harmonic analysis involves identifying special orthonormal bases for this vector space. Studying the behavior of function $f$ for points that have roughly $p$-fraction of the coordinates 1 is possible by considering the distribution of $p$-*biased* points $x \in \{\pm 1\}^N$, where $x$ is viewed as a random variable whose each coordinate is independently set to $+1$ with probability $p$. We denote this distribution as $\mathcal{B}_p$. Properties of functions $f$ on the hypercube in this setting are best studied by decomposing it using an *orthonormal basis* functions for the inner product space defined by $\langle \cdot, \cdot \rangle_{\mathcal{D}}$. It turns out that the functions $\{\phi_S : S \subseteq [N]\}$ defined as $\phi_S(x) = \prod_{i \in S} \left( \frac{x_i - \mu}{\sigma} \right)$ form the necessary orthonormal basis, where $\mu = 2p - 1$ and $\sigma^2 = 4p(1 - p)$ are the mean and variance of each coordinate of $x$. For orthonormality, we require that $\langle \phi_S, \phi_{S'} \rangle_{\mathcal{B}_p} = \mathbb{1}_{S=S'}$. To see why this is true, we note that

$$\langle \phi_S, \phi_{S'} \rangle_{\mathcal{B}_p} = \mathbb{E}_x \left[ \prod_{i \in S} \left( \frac{x_i - \mu}{\sigma} \right) \prod_{i \in S'} \left( \frac{x_i - \mu}{\sigma} \right) \right] \tag{11}$$

$$= \mathbb{E}_x \left[ \prod_{i \in S \cap S'} \left( \frac{x_i - \mu}{\sigma} \right)^2 \prod_{i \in S \Delta S'} \left( \frac{x_i - \mu}{\sigma} \right) \right] \tag{12}$$

$$= \prod_{i \in S \cap S'} \mathbb{E}_{x_i} \left[ \left( \frac{x_i - \mu}{\sigma} \right)^2 \right] \prod_{i \in S \Delta S'} \mathbb{E}_{x_i} \left[ \left( \frac{x_i - \mu}{\sigma} \right) \right] \tag{13}$$

$$= \prod_{i \in S \Delta S'} \mathbb{E}_{x_i} \left[ \left( \frac{x_i - \mu}{\sigma} \right) \right] \tag{14}$$

where the $S \Delta S'$ denotes the symmetric difference between the sets. The second to last equality follows from the independece of $x_i$'s in $\mathcal{B}_p$, while the last equality follows from the definition of mean $\mu$ and variance $\sigma^2$. Thus if $S = S'$, $S \Delta S' = \emptyset$ and this inner product is 1, while if $S \neq S'$, $|S \Delta S'| > 0$ and this product is 0 by definition of $\mu$. Hence $\phi_S$'s indeed form an orthonormal basis.

Given an orthonormal basis, every function $f\colon [N] \to \mathbb{R}$ can be expressed as a linear combination of this basis, i.e. $f = \sum_{S \subseteq [N]} \widehat{f}_S \phi_S$. The coefficients are often refered to $\widehat{f}_S$'s as "Fourier" coefficients of $f$, when the orthonormal basis is clear from context. The orthonormality of $\phi_S$'s also implies Parseval's identity: $\sum_S \widehat{f}_S^2 = \|f\|_{\mathcal{B}_p}^2$, since the $\ell_2$ norm measured using any orthonormal basis should be the same.

## A.2  LINEAR DATAMODELS

We provide some more background on linear datamodels from Ilyas et al. (2022) in the context of this paper; please refer to the original paper for more details. Consider a training set that is a subset of $[N]$, denoted by $x \in \{\pm 1\}^N$. The goal of datamodels is to be able to predict the outcome of training a neural network, using a fixed training algorithm, on a particular training set. Consider a function $f(x)$ that denotes that results of training on $x$, where $f$ can denote any function of the resultant network, e.g. test loss, loss on a particular test example, margin for a test example,

norm of parameters etc. Since the training algorithm involves randomness from various sources (initialization, SGD), $f$ can be viewed as an average over this randomness for a given training set $x$.

In general, $f$ could be a very complicated function of the training set $x$; for instance it could strongly depend on the presence or absence of particular subsets of datapoints. Furthermore due to the black-box nature of trained networks, one would not expect there to be a simple description of the function $f$. Rather surprisingly, Ilyas et al. (2022) found that for the function $f$ that denotes the margin for a single test example, $f$ can *on average* be approximated by a linear function of the indicator of the training set $x$, i.e. $f(x) \approx \theta^\top x$ for $x \sim \mathcal{B}_p$. One implication of this result is that the effect of adding or deleting a single point $i \in [N]$ is independent of the rest of the training set and only depends on the size of the training set. A more important consequence is that $\theta$ can be estimated by training evaluating $f$ on many $x$'s – which would require training a net for these $x$'s – and then fitting a linear function $\hat{\theta}$ on those samples. This can be used to not only estimate the individual influence $\hat{\theta}_i$ of datapoints, but also gives a way to predict $f$ on a new $x$ by using the estimate $\hat{\theta}^\top x$ rather than actually training a net on $x$. Details on how $\hat{\theta}$ is learned using Lasso regression (to encourage sparsity) can be found in Ilyas et al. (2022). Using Harmonic analysis, we show in Theorem 2.2 that the outcome of linear/Ridge/Lasso regression will provably be equal to the individual discrete influence of each point $i$.

## B  OMITTED PROOFS

### B.1  PROOFS FOR SECTION 2

We recall Theorem 2.2:

**Theorem 2.2** (Characterizing solutions to linear datamodels)**.** *Denote the quality of a linear datamodel $\theta \in \mathbb{R}^{N+1}$ on the $p$-biased distribution over training sets $\mathcal{B}_p$ by*

$$\mathcal{R}(\theta) := \mathbb{E}_{x \sim \mathcal{B}_p} \left[ \left( f(x) - \theta_0 - \sum_{i=1}^{N} \theta_i \bar{x}_i \right)^2 \right]. \tag{2}$$

*where $\bar{x} \in \{0, 1\}^N$ is the binary version for any $x \in \{\pm 1\}^N$. Then for $\mu = 2p - 1$ and $\sigma = \sqrt{4p(1-p)}$, the following are true about the optimal datamodels with and without regularization:*

(a) *The unregularized minimizer $\theta^\star = \arg\min \mathcal{R}(\theta)$ satisfies*

$$\theta_i^\star = \frac{2}{\sigma} \widehat{f}_{\{i\}}, \quad \theta_0^\star = \widehat{f}_\emptyset - \frac{(\mu+1)}{2} \sum_i \theta_i^\star. \tag{3}$$

*Furthermore the residual error is the sum of all Fourier coefficients of order 2 or higher.*

$$\mathcal{R}(\theta^\star) = B_{\geq 2} := \sum_{S \subseteq [N]:|S| \geq 2} \widehat{f}_S^2 \tag{4}$$

(b) *The minimizer with $\ell_2$ regularization $\theta^\star(\lambda, \ell_2) = \arg\min \left\{ \mathcal{R}(\theta) + \lambda \|\theta_{1:N}\|_2^2 \right\}$ satisfies*

$$\theta^\star(\lambda, \ell_2)_i = \frac{2}{\sigma} \left( 1 + \frac{4\lambda}{\sigma^2} \right)^{-1} \widehat{f}_{\{i\}}, \tag{5}$$

(c) *The minimizer with $\ell_1$ regularization $\theta^\star(\lambda, \ell_1) = \arg\min \left\{ \mathcal{R}(\theta) + \lambda \|\theta_{1:N}\|_1 \right\}$ satisfies*

$$\theta^\star(\lambda, \ell_1)_i = \frac{2}{\sigma} \left( \left( \widehat{f}_{\{i\}} - \lambda/\sigma \right)_+ - \left( -\widehat{f}_{\{i\}} - \lambda/\sigma \right)_+ \right) = \frac{2}{\sigma} \operatorname{sign}\left( \widehat{f}_{\{i\}} \right) \left( \left| \widehat{f}_{\{i\}} \right| - \lambda/\sigma \right)_+ \tag{6}$$

*where $z_+ = z \mathbb{1}_{z>0}$ is the standard ReLU operation.*

*Proof.* For all three parts, we will study the problem in the Fourier basis corresponding to the distribution $\mathcal{B}_p$ and the proof works for every $p \in (0, 1)$. Since $\{\phi_S\}_{S \subseteq [N]}$ is an orthonormal basis for the inner product space with $\langle f, g \rangle_{\mathcal{B}_p} = \mathop{\mathbb{E}}_{x \sim \mathcal{B}_p} [f(x)g(x)]$, we can rewrite $\mathcal{R}(\theta)$ as follows

$$\mathcal{R}(\theta) := \mathbb{E}_{x \sim \mathcal{B}_p} \left( f(x) - \theta_0 - \sum_i \theta_i \bar{x}_i \right)^2 =^{(i)} \mathbb{E}_{x \sim \mathcal{B}_p} \left( f(x) - \bar{\theta}_0 - \sum_i \bar{\theta}_i \phi_{\{i\}}(x) \right)^2 \tag{15}$$

$$=^{(ii)} \left\| f - \bar{\theta}_0 - \sum_i \bar{\theta}_i \phi_{\{i\}} \right\|_{\mathcal{B}_p}^2, \quad \text{where } \|g\|_{\mathcal{B}_p} := \langle g, g \rangle_{\mathcal{B}_p} = \mathbb{E}_{x \sim \mathcal{B}_p}[g(x)^2] \tag{16}$$

$$\text{and } \bar{\theta}_i = \frac{\sigma}{2} \theta_i, \quad \bar{\theta}_0 = \theta_0 + \frac{(\mu + 1)}{2} \sum_i \theta_i \tag{17}$$

The first equality $(i)$ follows by observing that $x_i = 2\bar{x}_i - 1$ and that $\phi_{\{i\}}(x) = (x_i - \mu)/\sigma$. Step $(ii)$ follows by applying the definition of $\|g\|_{\mathcal{B}_p}$ to the function $g = f - \bar{\theta}_0 - \sum_i \bar{\theta}_i \phi_{\{i\}}$.

We can further simplify Equation (16) as follows:

$$\mathcal{R}(\theta) = \left\| f - \bar{\theta}_0 - \sum_i \bar{\theta}_i \phi_{\{i\}} \right\|_{\mathcal{B}_p}^2 = \left\| \hat{f}_\emptyset - \bar{\theta}_0 + \sum_{S:|S|=1} \left( \hat{f}_S - \bar{\theta}_i \right) \phi_S + \sum_{S:|S|\geq 2} \hat{f}_S \phi_S \right\|_{\mathcal{B}_p}^2 \tag{18}$$

$$=^{(i)} \left( \bar{\theta}_0 - \hat{f}_\emptyset \right)^2 + \sum_{i=1}^{N} \left( \bar{\theta}_i - \hat{f}_{\{i\}} \right)^2 + \sum_{S:|S|\geq 2} \hat{f}_S^2 \tag{19}$$

where $(i)$ uses orthonormality of $\{\phi_S\}_{S \subseteq [N]}$ and Parseval's theorem (O'Donnell, 2014). With this expression for $\mathcal{R}(\theta)$, we are now ready to prove the main results.

**Proof for (a)**: From Equation (19), it is evident that the minimizer for the unregularized objective will satisfy $\hat{\theta}_0^\star = \hat{f}_\emptyset$ and $\bar{\theta}_i^\star = \hat{f}_{\{i\}}$ for $i \in [N]$. Plugging this into Equation (17) yields the desired expressions for $\theta^\star$. Furthermore, the residual for this optimal $\theta^\star$ is $\sum_{S:|S|\geq 2} \hat{f}_S^2$.

**Proof for (b)**: The $\ell_2$ regularized objective can be written as follows:

$$\mathcal{R}(\theta) + \lambda \|\theta_{1:N}\|_2^2 = \mathcal{R}(\theta) + \lambda \sum_{i=1}^{N} \theta_i^2 =^{(i)} \mathcal{R}(\theta) + \frac{4\lambda}{\sigma^2} \sum_{i=1}^{N} \bar{\theta}_i^2 \tag{20}$$

where $(i)$ follows from Equation (17). Combining this with Equation (19), we observe that minimization w.r.t. $\bar{\theta}_i$ can be done independently of each other. Thus $\bar{\theta}_i^\star = \arg\min_{\bar{\theta}_i} \left( \bar{\theta}_i - \hat{f}_{\{i\}} \right)^2 + \frac{4\lambda}{\sigma^2} \bar{\theta}_i^2 = \left( 1 + \frac{4\lambda}{\sigma^2} \right)^{-1} \hat{f}_{\{i\}}$. Plugging this into Equation (17) gives the desired expression for $\theta^\star$.

**Proof for (c)**: The $\ell_1$ regularized objective can be written as follows:

$$\mathcal{R}(\theta) + \lambda \|\theta_{1:N}\|^1 = \mathcal{R}(\theta) + \lambda \sum_{i=1}^{N} |\theta_i| =^{(i)} \mathcal{R}(\theta) + \frac{2\lambda}{\sigma} \sum_{i=1}^{N} |\bar{\theta}_i| \tag{21}$$

Again the minimization w.r.t. $\bar{\theta}_i$ can be done independently: $\bar{\theta}_i^\star = \arg\min_{\bar{\theta}_i} \left( \bar{\theta}_i - \hat{f}_{\{i\}} \right)^2 + \frac{2\lambda}{\sigma} |\bar{\theta}_i| = \left( \hat{f}_{\{i\}} - \lambda/\sigma \right)_+ - \left( -\hat{f}_{\{i\}} - \lambda/\sigma \right)_+$. Plugging this into Equation (17) gives the desired expression for $\theta^\star$. $\qquad \square$

## B.2 Proofs for Section 3

We recall Theorem 3.1

**Theorem 3.1.** *The quality of the best linear approximation to $f$ can be bounded in terms of the normalized noise stability $\bar{h}$ as*

$$\textit{(Normalized residual)} \quad \frac{\sum_{S:|S|\geq 2} \hat{f}_S^2}{\sum_S \hat{f}_S^2} \leq \frac{1 - \bar{h}(\rho)}{1 - \rho^2}. \tag{8}$$

*Proof.* We first note that $\bar{h}(\rho) = h(\rho)/h(1)$ since $\mathbb{E}[f(x)^2]$ is the noise stability when $x' = x$ which happens at $\rho = 1$. Let $B_k = \sum_{S:|S|=k} \widehat{f}_S^2$. From Equation (7) we get

$$\bar{h}(\rho) = \frac{\sum_S \rho^{|S|} \widehat{f}_S^2}{\sum_S \widehat{f}_S^2} = \frac{\sum_k \rho^k B_k}{\sum_k B_k} \leq \frac{B_0 + B_1 + \rho^2 \sum_{k \geq 2} B_k}{\sum_k B_k}$$

Let $B = \sum_S \widehat{f}_S^2 = \sum_k \widehat{f}_k^2$ and $B_{\geq 2} = \sum_{S:|S| \geq 2} \widehat{f}_S^2 = \sum_{k \geq 2} \widehat{f}_k^2$. Then we get,

$$\bar{h}(\rho) \leq \frac{(B - B_{\geq 2}) + \rho^2 B_{\geq 2}}{B} = 1 - (1 - \rho^2)\frac{B_{\geq 2}}{B} \implies \frac{B_{\geq 2}}{B} \leq \frac{1 - \bar{h}(\rho)}{1 - \rho^2}$$

This completes the proof. $\qquad\square$

**Lemma B.1.** *For a function $f$ and $\rho \in [0,1]$ and evaluation budget $n$, let $\hat{h}(\rho) = \text{NOISESTABILITY}(f, n, \rho)$ be the noise stability estimate from Algorithm 1. If $|f(x)| \leq C$ for every $x$, then with probability at least $1 - \eta$, the error in the estimate can be upper bounded by*

$$\left|\hat{h}(\rho) - h(\rho)\right| \leq \delta(n) = \mathcal{O}\left(\sqrt{C^2 \log(1/\eta)/n}\right) \tag{22}$$

*Proof.* The proof follows from a straightforward application of Hoeffding's inequality. Note that $\hat{h}(\rho) = \frac{2}{n} \sum_{i=1}^{n/2} [f(x^{(i)})f(x'^{(i)})]$ is a sum of $n/2$ i.i.d. variables that are all in the range $[-C^2, C^2]$. Additionally since $\mathbb{E}[\hat{h}(\rho)] = h(\rho)$, Hoeffding's inequality gives us

$$P\left(\left|\hat{h}(\rho) - h(\rho)\right| \geq \delta\right) \leq 2e^{-\frac{\delta^2 n}{C^2}}. \tag{23}$$

Setting $\delta = \sqrt{C^2 \log(2/\eta)/n}$ makes this probability at most $\eta$, thus completing the proof. $\qquad\square$

We now prove Theorem 3.2. Recall the statement:

**Theorem 3.2.** *Let $\hat{B}_{\geq 2} = \text{RESIDUALESTIMATION}(f, n, [0, \rho, 2\rho], 2)$ be the estimated residual (see Algorithm 1) after fitting a degree 2 polynomial to noise stability estimates at $0, \rho, 2\rho$, using $n$ calls to $f$. If $n = \mathcal{O}(1/\epsilon^3)$ and $\rho = \sqrt{\epsilon}$, then with high probability we have that $|\hat{B}_{\geq 2} - B_{\geq 2}| \leq \epsilon$.*

*Proof.* Using standard concentration inequalities, Lemma B.1 shows that the estimations are close enough to the true values with high probability, i.e. $\boldsymbol{y}_i = \hat{h}(\rho_i) = h(\rho_i) + \delta_i$ where $|\delta_i| \leq \delta$ where $\delta := \delta(n/k) = \mathcal{O}\left(\sqrt{k/n}\right)$ from Lemma B.1. Let $\hat{\boldsymbol{z}}$ be the solution to Algorithm 1 and let $\boldsymbol{z}^\star = [B_j]_{j=0}^d$ be the "true" coefficients up to degree $d$. Since $\hat{\boldsymbol{z}}$ minimizes $\|\boldsymbol{A}\boldsymbol{z} - \boldsymbol{y}\|^2$ and $\boldsymbol{z}^\star$ is also a valid solution, we have

$$\|\boldsymbol{A}\hat{\boldsymbol{z}} - \boldsymbol{y}\|^2 \leq \|\boldsymbol{A}\boldsymbol{z}^\star - \boldsymbol{y}\|^2 \tag{24}$$

Furthermore, we note the following about $\boldsymbol{A}\boldsymbol{z}^\star$:

$$(\boldsymbol{A}\boldsymbol{z}^\star)_i = \sum_{j=0}^d B_j \rho_i^j = h(\rho_i) - \sum_{j > d} B_j \rho^i \in \left[h(\rho_i) - B_{>d}\, \rho_i^{d+1}, h(\rho_i)\right] \tag{25}$$

We can upper bound $\|\boldsymbol{A}\boldsymbol{z}^\star - \boldsymbol{y}\|^2$ by observing that

$$|(\boldsymbol{A}\boldsymbol{z}^\star)_i - \boldsymbol{y}_i| \leq |\delta_i| + B_{>d}\, \rho_i^{d+1} \leq \delta + B_{>d}\, \rho_i^{d+1} \tag{26}$$

$$\|\boldsymbol{A}\boldsymbol{z}^\star - \boldsymbol{y}\|^2 \leq k\left(\delta + B_{>d}\, \rho_i^{d+1}\right)^2 \tag{27}$$

Using these, we measure the closeness of $\hat{\boldsymbol{z}}$ to $\boldsymbol{z}^\star$ as follows:

$$\|\boldsymbol{A}\boldsymbol{z}^\star - \boldsymbol{A}\hat{\boldsymbol{z}}\|^2 \leq 2\left(\|\boldsymbol{A}\boldsymbol{z}^\star - \boldsymbol{y}\|^2 + \|\boldsymbol{A}\hat{\boldsymbol{z}} - \boldsymbol{y}\|^2\right) \leq 4\|\boldsymbol{A}\boldsymbol{z}^\star - \boldsymbol{y}\|^2 \tag{28}$$

where the first inequality follows from triangle inequality and Cauchy-Schwarz inequality, and the last inequality follows from Equation (24).

A naive upper bound for $\|\boldsymbol{z}^\star - \hat{\boldsymbol{z}}\|$ is $\lambda_{\min}(\boldsymbol{A})^{-1}\|\boldsymbol{A}\boldsymbol{z}^\star - \boldsymbol{A}\hat{\boldsymbol{z}}\|$. However this turns out to be quite a large and suboptimal upper bound. Since Algorithm 1 only returns $\hat{\boldsymbol{z}}_1$ and $\hat{\boldsymbol{z}}_2$, corresponding to estimates of $B_0$ and $B_1$, we only need to upper bound $|\hat{\boldsymbol{z}}_i - \boldsymbol{z}_i^\star|$ for $i \in [2]$.

We now analyze the special case of $k = 3$, $d = 2$, $\rho_1 = 0, \rho_2 = \rho, \rho_3 = 2\rho$. Here we have $\boldsymbol{A} = \begin{pmatrix} 1 & 0 & 0 \\ 1 & \rho & \rho^2 \\ 1 & 2\rho & 4\rho^2 \end{pmatrix}$. Let $\Delta = \boldsymbol{z}^\star - \hat{\boldsymbol{z}}$. Firstly note that $(\boldsymbol{A}\Delta)_1 = \Delta_1$ and so $|\Delta_1| \leq \|\boldsymbol{A}\Delta\|$. To upper bound $\Delta_2$, we note that

$$\boldsymbol{A}\Delta = \Delta_2 \begin{pmatrix} 0 \\ \rho \\ 2\rho \end{pmatrix} + \begin{pmatrix} 1 & 0 \\ 1 & \rho^2 \\ 1 & 4\rho^2 \end{pmatrix} \begin{pmatrix} \Delta_1 \\ \Delta_3 \end{pmatrix} = \rho\Delta_2 \underbrace{\begin{pmatrix} 0 \\ 1 \\ 2 \end{pmatrix}}_{\boldsymbol{v}} + \underbrace{\begin{pmatrix} 1 & 0 \\ 1 & 1 \\ 1 & 4 \end{pmatrix}}_{\boldsymbol{B}} \underbrace{\begin{pmatrix} \Delta_1 \\ \rho^2\Delta_3 \end{pmatrix}}_{\boldsymbol{u}} \tag{29}$$

$$\|\boldsymbol{A}\Delta\| = \|\rho\Delta_2\boldsymbol{v} + \boldsymbol{B}\boldsymbol{u}\| \geq \min_{\boldsymbol{w}} \|\rho\Delta_2\boldsymbol{v} + \boldsymbol{B}\boldsymbol{w}\| = \rho|\Delta_2| \left\|\left(I - \boldsymbol{B}^\dagger \boldsymbol{B}\right)\boldsymbol{v}\right\| \geq 0.39\rho|\Delta_2| \tag{30}$$

where the last inequality follows from numeric calculation of the norm, and the last inequality follows from the fact that the minimum $\ell_2$ can be obtained by finding the residual after projecting onto $\boldsymbol{B}$. Thus $|\hat{\boldsymbol{z}}_1 - B_0| = |\Delta_1| \leq \|\boldsymbol{A}\Delta\|$ and $|\hat{\boldsymbol{z}}_2 - B_1| = |\Delta_1| \leq \|\boldsymbol{A}\Delta\|/(0.39\rho)$.

Combining Equations (27), (28) and (30), we get that

$$|\hat{\boldsymbol{z}}_1 - B_0| \text{ and } |\hat{\boldsymbol{z}}_2 - B_1| \leq \mathcal{O}\left(\frac{\delta + B_{\geq 3}\rho^3}{\rho}\right) = \mathcal{O}\left(\frac{\delta}{\rho} + B_{\geq 3}\rho^2\right) \tag{31}$$

Picking the optimal value of $\rho = \Theta\left((\delta/B_{\geq 3})^{1/3}\right) = \mathcal{O}\left(n^{-1/6}B_{\geq 3}^{-1/3}\right)$, and using the fact that $\hat{B}_{\geq 2} = \hat{h}(1) - \hat{\boldsymbol{z}}_1 - \hat{\boldsymbol{z}}_2$ we get

$$B_{\geq 2} = h(1) - B_0 - B_1 = \hat{B}_{\geq 2} + \mathcal{O}\left(\delta^{2/3}B_{\geq 3}^{1/3}\right) = \tilde{\mathcal{O}}\left(\left(\frac{B_{\geq 3}}{n}\right)^{1/3}\right) \tag{32}$$

Thus to achieve an $\epsilon$ approximation, we need $\rho = \Theta\left(\sqrt{\epsilon}\right)$ and $n = \Omega\left(B_{\geq 3}/\epsilon^3\right)$, which completes the proof.

$\square$

## B.3 Proofs for Section 4

We recall Theorem 4.1

**Theorem 4.1.** *[Group influence] The following are true about group influence of deletion.*

$$Inf_I(f) := \mathbb{E}_{x \sim \mathcal{B}_p}\left[f(x) - f(x|_{I \to -1})\right] = p\sum_{i \in I} \theta^\star - \sum_{I' \subseteq I, |I'| \geq 2} (-1)^{|I'|}\left(\frac{p}{1-p}\right)^{|I'|/2} \widehat{f}_{I'}$$

$$= p\sum_{i \in I} Inf_i(f) - \sum_{I' \subseteq I, |I'| \geq 2} (-1)^{|I'|}\left(\frac{p}{1-p}\right)^{|I'|/2} \widehat{f}_{I'}$$

*where $\theta^\star$ is the optimal datamodel (Equation (3)) and $Inf_i$ is an individual influence (Equation (1)).*

*Proof.* Recall that the function $f$ can be decomposed into the Fourier basis as $f(x) = \sum_{S \subseteq [N]} \widehat{f}_S \phi_S(x) = \sum_{S \subseteq [N]} \widehat{f}_S \prod_{i \in S} \frac{(x_i - \mu)}{\sigma}$, where $\mu = 2p - 1$ and $\sigma = \sqrt{4p(1-p)}$. Thus setting $x_I$ to $-1$ will have the following effect:

$$f(x|_{I \to -1}) = \sum_{S \subseteq [N]} \widehat{f}_S \left(\frac{-1 - \mu}{\sigma}\right)^{|S \cap I|} \phi_{S \setminus I}(x) = \sum_{S \subseteq [N]} \widehat{f}_S \left(\frac{-2p}{2\sqrt{p(1-p)}}\right)^{|S \cap I|} \phi_{S \setminus I}(x)$$

$$\tag{33}$$

$$= \sum_{S \subseteq [N]} \widehat{f}_S (-1)^{|S \cap I|} \left( \frac{p}{1-p} \right)^{|S \cap I|/2} \phi_{S \setminus I}(x) \tag{34}$$

$$=^{(i)} \sum_{S : S \cap I = \emptyset} \phi_S(x) \sum_{I' \subseteq I} (-1)^{|I'|} \left( \frac{p}{1-p} \right)^{|I'|/2} \widehat{f}_{S \cup I'} \tag{35}$$

where in step $(i)$ we collect all terms that share the same $\phi_{S \setminus I}$ by relabeling $S \leftarrow S \setminus I$ and $I' \leftarrow S \cap I$. This proves the first part of the result.

For the second part we first note that $\mathbb{E}_x[f(x)] = \widehat{f}_\emptyset$. Secondly, the only term that remains in $\mathbb{E}_x[f(x|_{I \to -1})]$ is the constant term (i.e. coefficient of the basis function $\phi_\emptyset(x)$. From Equation (35), only the terms with $S = \emptyset$ remain. So,

$$\mathbb{E}_x \left[ f(x) - f(x|_{I \to -1}) \right] = \widehat{f}_\emptyset - \sum_{I' \subseteq I} (-1)^{|I'|} \left( \frac{p}{1-p} \right)^{|I'|/2} \widehat{f}_{I'} \tag{36}$$

$$=^{(i)} \sum_{i \in I} \sqrt{\frac{p}{1-p}} \widehat{f}_{\{i\}} - \sum_{I' \subseteq I, |I'| \geq 2} (-1)^{|I'|} \left( \frac{p}{1-p} \right)^{|I'|/2} \widehat{f}_{I'} \tag{37}$$

$$= p \sum_{i \in I} \frac{2}{\sigma} \widehat{f}_{\{i\}} - \sum_{I' \subseteq I, |I'| \geq 2} (-1)^{|I'|} \left( \frac{p}{1-p} \right)^{|I'|/2} \widehat{f}_{I'} \tag{38}$$

$$=^{(ii)} p \sum_{i \in I} \theta_i^\star - \sum_{I' \subseteq I, |I'| \geq 2} (-1)^{|I'|} \left( \frac{p}{1-p} \right)^{|I'|/2} \widehat{f}_{I'} \tag{39}$$

$$=^{(iii)} p \sum_{i \in I} \text{Inf}_i(f) - \sum_{I' \subseteq I, |I'| \geq 2} (-1)^{|I'|} \left( \frac{p}{1-p} \right)^{|I'|/2} \widehat{f}_{I'} \tag{40}$$

where $(i)$ follows by separating out the size 0 and size 1 $I'$'s, $(ii)$ follows from Theorem 2.2 and $(iii)$ follows from Proposition 2.1.

$\square$

We now show an upper bound on the residual term from the previous result in terms of the residual of a linear datamodels.

**Lemma B.2.** *The residual term of group influence minus sum of individual influences from Theorem 4.1 can be upper bounded as follows:*

$$\left| \sum_{I' \subseteq I, |I'| \geq 2} (-1)^{|I'|} \left( \frac{p}{1-p} \right)^{|I'|/2} \widehat{f}_{I'} \right| < (1-p)^{-|I|/2} \sqrt{B_{\geq 2}} \tag{41}$$

*where $B_{\geq 2}$ is the residual of the best linear datamodel as defined in Equation (4).*

*Proof.* We use Cauchy-Schwarz inequality to upper bound this residual.

$$\left( \sum_{I' \subseteq I, |I'| \geq 2} (-1)^{|I'|} \left( \frac{p}{1-p} \right)^{|I'|/2} \widehat{f}_{I'} \right)^2 \leq \left( \sum_{I' \subseteq I, |I'| \geq 2} \left( \frac{p}{1-p} \right)^{|I'|} \right) \left( \sum_{I' \subseteq I, |I'| \geq 2} \widehat{f}_{I'}^2 \right) \tag{42}$$

$$< \left( \sum_{I' \subseteq I} \left( \frac{p}{1-p} \right)^{|I'|} \right) \left( \sum_{I' \subseteq [N], |I'| \geq 2} \widehat{f}_{I'}^2 \right) \tag{43}$$

$$= \left( 1 + \frac{p}{1-p} \right)^{|I|} B_{\geq 2} = (1-p)^{-|I|} B_{\geq 2} \tag{44}$$

This completes the proof.

$\square$

We now prove the exponential blow up in group influence from Lemma 4.2 (restated below).

**Lemma 4.2.** *Consider the function[5] $f$ from Example 1 with $\beta = 0.5$ and $\alpha \to \infty$; so $f(x) = \mathbb{1}\{avg(x_A) > 0.5\}$. If $x$ is $p$-biased with $p > 0.5$, then for any constant fraction subset $A' \subseteq A$, its group influence is exponentially larger than sum of individual influences.*

$$Inf_{A'}(f) \geq \frac{(2(1-p))^{-|A'|+1}}{|A'|} \left( \sum_{i \in A'} Inf_i(f) \right). \tag{9}$$

*Proof.* Let $M = |A|$ and suppose $|A'| = \gamma|A|$ for a small constant $\gamma$. We study $\text{Inf}_{A'}(f)$ using its definition and for a general $\beta \in (0,1)$. The function of interest is $f(x) = \mathbb{1}\{avg(x_A) > \beta\}$. Let $B(n,q)$ denote the binomial r.v. that is a sum of $n$ independent Bernoulli's with parameter $q$. We first note that we can rewrite the expected value of $f$ as follows:

$$\mathbb{E}_x[f(x)] = \Pr(B(M,p) > \beta M) \tag{45}$$

This is because the only times $f(x)$ is 1 is when at least $\beta$ fraction of indices in $A$ are 1, which is precisely capture by the Binomial distribution tail probability. Similarly, we can argue that

$$\mathbb{E}_x[f(x|_{A' \to -1})] = \Pr(B(M(1-\gamma),p) > \beta M) \tag{46}$$

The group influence can be calculated as follows:

$$\text{Inf}_{A'}(f) = \mathbb{E}_x[f(x)] - \mathbb{E}_x[f(x|_{A' \to -1})] \tag{47}$$

$$= \Pr(B(M,p) > \beta M) - \Pr(B(M(1-\gamma),p) > \beta M) \tag{48}$$

$$= \Pr(B(M(1-\gamma),p) \leq \beta M) - \Pr(B(M,p) \leq \beta M) \tag{49}$$

$$= \sum_{\beta' \in [\frac{1}{M}, \frac{2}{M}, \dots \beta]} \binom{M(1-\gamma)}{\beta' M} p^{\beta' M}(1-p)^{M(1-\gamma-\beta')} \tag{50}$$

$$- \sum_{\beta' \in [\frac{1}{M}, \frac{2}{M}, \dots \beta]} \binom{M}{\beta' M} p^{\beta' M}(1-p)^{M(1-\beta')} \tag{51}$$

Define $c_\beta(\gamma) = \binom{M(1-\gamma)}{\beta M} p^{\beta M}(1-p)^{M(1-\gamma-\beta)}$. The the above expression reduces to

$$\text{Inf}_{A'}(f) = \sum_{\beta' \in [\frac{1}{M}, \frac{2}{M}, \dots \beta]} (c_{\beta'}(\gamma) - c_{\beta'}(0)). \tag{52}$$

Note that individual influences $\text{Inf}_i(f)$ corresponds sets of size $|A'| = 1$, i.e. $\gamma = 1/M$. So,

$$\text{Inf}_i(f) = \sum_{\beta' \in [\frac{1}{M}, \frac{2}{M}, \dots \beta]} (c_{\beta'}(1/M) - c_{\beta'}(0)). \tag{53}$$

We study the quantity $c_\beta(\gamma)$ in more detail. In particular, we use Stirling's approximation for binomial coefficients $\log_2 \binom{n}{qn} \approx nH(q)$, where $H(q) = -q \log_2 q - (1-q) \log_2(1-q)$ is the entropy function. This gives a cleaner expression for $\log_2(c_\beta(\gamma))$.

$$\log_2(c_\beta(\gamma)) = \log_2 \binom{M(1-\gamma)}{\beta M} + \beta M \log_2(p) + (1-\beta-\gamma) \log_2(1-p) \tag{54}$$

$$= M(1-\gamma)H\left(\frac{\beta}{1-\gamma}\right) + M\beta \log_2(p) + M(1-\beta-\gamma) \log_2(1-p) \tag{55}$$

For a small $\gamma$, we use a linear approximation for $\log_2(c_\beta(\gamma))$ at $\gamma = 0$. The derivate is

$$\frac{d}{d\gamma} \log_2(c_\beta(\gamma))|_{\gamma=0} = M\left(\log_2\left(\frac{1-\beta}{\beta}\right) - H(\beta) - \log_2(1-p)\right) = g(\beta) \tag{56}$$

where $g(\beta)$ is a decreasing function of $\beta$. This gives us the approximation for $c_\beta(\gamma)$

$$\log_2(c_\beta(\gamma)) \approx \log_2(c_\beta(0)) + \gamma g(\beta) \tag{57}$$

---

[5]The result can potentially be shown for more general $\alpha, \beta, p$

$$c_\beta(\gamma) \approx c_\beta(0) 2^{\gamma g(\beta)} \tag{58}$$

We note that for $\beta' \leq 0.5$ and $p > 1/2$, $g(\beta') > 0$ and so $c_{\beta'}(\gamma)$ is an increasing function in $\gamma$. For ratio of group to individual influence from Equations (52) and (53), we consider the following expression for $\beta' \leq \beta = 0.5$:

$$\frac{c_{\beta'}(\gamma) - c_{\beta'}(0)}{c_{\beta'}(1/M) - c_{\beta'}(0)} \geq \frac{c_{\beta'}(\gamma)}{c_{\beta'}(1/M)} \approx 2^{(\gamma - 1/M)g(\beta')} \geq 2^{(\gamma - 1/M)g(0.5)} \tag{59}$$

$$= 2^{(\gamma - 1/M)M(-1 - \log(1-p))} = (2(1-p))^{-\gamma M + 1} \tag{60}$$

$$= (2(1-p))^{-|A'| + 1} \tag{61}$$

Plugging this back into Equations (52) and (53), we see that the terms for each $\beta'$ in the summation are upper bounded by Equation (61). This gives

$$\frac{\mathrm{Inf}_{A'}(f)}{\mathrm{Inf}_i(f)} \geq (2(1-p))^{-|A'| + 1} \tag{62}$$

Considering the total individual influences instead of just influence for just 1 point completes the proof. $\square$

## C EXPERIMENTAL SETUP

Experiments are conducted on the CIFAR-10 data to test the estimation procedure and the quality of the linear fit in Figures 1 and 2. We use the FFCV library Leclerc et al. (2022) to train models on CIFAR-10; each model takes $\sim 30$s to train on our GPUs. We first pick subset of 10k images from the CIFAR-10 training dataset. Our models trained are then trained on sets of size 5000 (corresponding to $p = 0.5$), which achieve an average of $\sim 71\%$ accuracy on the CIFAR-10 test set. For the noise stability estimates from Algorithm 1, we sample 600 pairs of $\rho$-correlated datasets $(x, x')$ each for $\rho = 0.05, 0.1$, and $0.2$. We train 12,000 models for each setting of $\rho$, where there are 600 distinct sets $x$ + 600 distinct $\rho$-correlated sets $x'$ chosen and the remaining $10\times$ runs are due to running 10 random seeds per training set. We use the default ResNet based architecture in FFCV with a batch size of 512, an initial learning rate of 0.5, 24 epochs, weight decay of $5e\text{-}4$ and SGD with momentum as the optimizer.

For the experiment with residual estimation, we use the set of $\rho$'s to be $[0, 0.1, 0.2, 1]$ for most experiments, $[0, 0.05, 0.1, 1]$ for Figure 2c and $[0, 0.1, 0.2]$ for Figure 4. Furthermore we use degree $d = 2$ in Algorithm 1 for most experiments and $d = 3$ for comparison in Figure 2b. Although the theoretical analysis in Theorem 3.2 does not use $\rho = 1$ for the polynomial fitting, we find experimentally that adding $\rho = 1$ gives more robust estimations. This is evident from the polynomial fits obtained for 20 randomly selected test examples in Figures 3 and 4, where the fits from $[0, 0.1, 0.2, 1]$ are clearly better than those from $[0, 0.1, 0.2]$. The theory can also be extended for this case, with a slightly modified analysis.

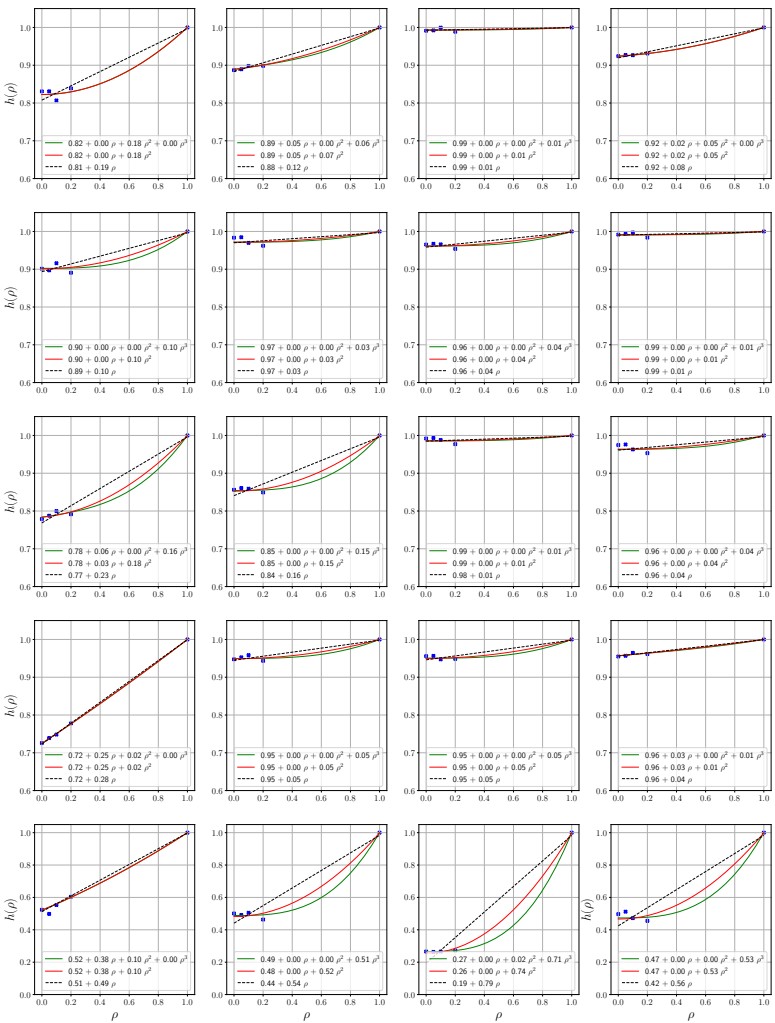

Figure 3: Noise stability estimates plotted for 20 randomly sampled test examples. For each example (corresponding to a plot), we plot the estimated $\hat{h}(\rho)$ in blue dots and the best degree $d \in \{1, 2, 3\}$ (black, red and blue curves respectively) polynomial fits obtained from Algorithm 1 for the list of $\rho$'s being $[0, 0.1, 0.2, 1]$. The test examples are rearranged such that the first 4 rows contain points with $\min_\rho \hat{h}(\rho) > 0.7$ and the y-axis range is set to $[0.6, 1]$ for clarity of presentation. The y-axis range for the last row is set to $[0, 1]$. The estimated residual from Algorithm 1 is precisely equal to one minus the sum of the constant and degree-1 coefficients.

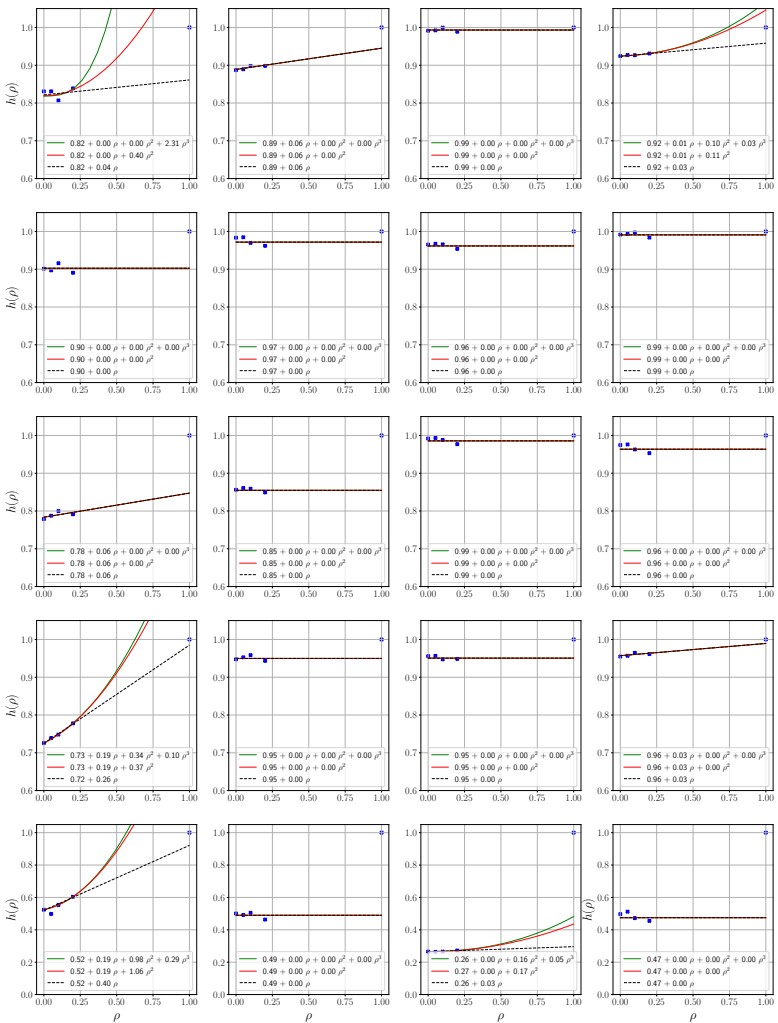

Figure 4: Noise stability estimates plotted for 20 randomly sampled test examples. For each example (corresponding to a plot), we plot the estimated $\hat{h}(\rho)$ in blue dots and the best degree $d \in \{1, 2, 3\}$ (black, red and blue curves respectively) polynomial fits obtained from Algorithm 1 for the list of $\rho$'s being $[0, 0.1, 0.2]$. The test examples are rearranged such that the first 4 rows contain points with $\min_\rho \hat{h}(\rho) > 0.7$ and the y-axis range is set to $[0.6, 1]$ for clarity of presentation. The y-axis range for the last row is set to $[0, 1]$. The estimated residual from Algorithm 1 is precisely equal to one minus the sum of the constant and degree-1 coefficients.

