# OpenReview forum: "Understanding Influence Functions and Datamodels via Harmonic Analysis"
_ICLR.cc/2023/Conference — ICLR 2023 poster_

### Official Review · Reviewer_cauj · 2022-10-21

**Confidence:** 3
**Correctness:** 4
**Technical Novelty And Significance:** 4
**Empirical Novelty And Significance:** 3
**Recommendation:** 8

**Clarity, Quality, Novelty And Reproducibility:**

### Clarity
I think the paper is written clearly and the main idea and the results are easy to follow.
As a minor suggestion, I think the paper becomes much easier for the readers if there is a brief overview of the linear datamodel.
The linear datamodel is not that popular compared to influence function.

### Quality, Novelty
The problem considered in this paper, characterizing the connection between influence function and the linear data model, is an interesting problem and the finding would be novel.
The finding explains the success of the linear datamodel.

### Reproducibility
The experimental setups are provided in appendix.
The results will be reproducible.

**Strength And Weaknesses:**

### Strength
The major strength of the paper is on the characterization of the connections between influence function and the linear datamodel.
This characterization successfully explained the finding of the linear datamodel.
The authors also found some counter cases when the connection fails.
The authors further provided an efficient way to estimate the fitting error so that one can assess whether the linear datamodel is valid on the dataset at hand.

### Weakness
I do not find any crucial weakness.
As a minor suggestion, I think the paper becomes much easier for the readers if there is a brief overview of the linear datamodel.
The linear datamodel is not that popular compared to influence function.

**Summary Of The Paper:**

This paper revealed the connection of influence function and the linear datamodel using harmonic analysis.
Linear datamodel is a model fitted with an $N$-dimensional binary vector as its input and with the outcome (or loss) of a model trained on a subset of $N$ training points indicated by the binary vector as its regression target.
A recent study observed that the influence function (more formally, discrete influence function) can be approximated by this linear datamodel.
This paper provides a unified view of this observation why such an approximation is possible.
Let $f$ be a training algorithm that maps the $N$ dimensional binary vector to the target.
For the analysis, the authors considered expanding $f$ using an orthonormal basis in $N$ dimensional space.
This expansion allows us to view $f$ as a function on the $N$ dimensional grid rather than the training algorithm.
With this expansion, we can see that discrete influence function corresponds to the coefficient of the degree-1 basis.
This finding explains why the linear datamodel can approximate discrete influence function.
If the coefficients of $f$ in the higher-degree basis are small, the basis of degree-1 can well approximate $f$.
The authors also analyzed a case when this approximation is possible by introducing the notion of noise stability.
The authors also provided an efficient algorithm for estimating the approximation error.


**Summary Of The Review:**

This paper provide the characterization of the connections between influence function and the linear data model.
This characterization successfully explained the finding of the linear datamodel.
This is an interesting problem and the finding would be novel.
The paper also provides some counter cases when the connection fails, and also an efficient way to estimate the fitting error so that one can assess whether the linear datamodel is valid on the dataset at hand.

---

> ### Author Response · Authors · 2022-11-14
> **Response to Reviewer cauj**
>
> We thank the reviewer for the positive review. The main suggestion about linear datamodels is addressed below.
>
> > As a minor suggestion, I think the paper becomes much easier for the readers if there is a brief overview of the linear datamodel. The linear datamodel is not that popular compared to influence function.
>
> Thank you for the suggestion. Besides what is in Section 2.3 (i.e., linear datamodels requires solving a Lasso regression problem), we will include some more details about linear datamodels and their applications in the main paper or the Appendix.

---

> > ### Comment · Reviewer_cauj · 2022-12-05
> > **Reply to authors**
> >
> > I would like to thank the authors for considering the paper update. I strongly believe that update will make the paper easier to read even for readers who are not familiar with datamodel.

---

### Official Review · Reviewer_rQ2x · 2022-10-23

**Confidence:** 3
**Clarity, Quality, Novelty And Reproducibility:** This paper is novel and interesting w…
**Correctness:** 3
**Technical Novelty And Significance:** 4
**Empirical Novelty And Significance:** 4
**Recommendation:** 6

**Strength And Weaknesses:**

Strength:
1. The research problem, analyzing the influence of data points, is very interesting and important for explainable AI.
2. The paper provides a theoretical analysis of the linear datamodel, and explained why the effects of number of points deleted is observed in practice as linear rather than exponential.
3. Experimental results verified the proposed method

Weaknesses:
1. This paper is not friendly to most beginners, and the main idea is hard to follow. The authors also need to explain more detailedly how the residual estimation can predict the performance.

**Summary Of The Paper:**

This paper focuses on a very interesting and important research topic, predicting the predictions from training data. Built on the datamodels proposed in [1], this paper seeks to provide a better theoretical understanding of why a linear regression method can predict the effect of training data. More importantly, this paper designs a new algorithm for estimating the approximated linear datamodel with much less training cost. This paper provides an exciting view to the counter-factual explanations of training data.

[1]Andrew Ilyas, Sung Min Park, Logan Engstrom, Guillaume Leclerc, and Aleksander Madry. Datamodels: Predicting predictions from training data, ICML 2022

**Summary Of The Review:**

This paper is generally well-written and novel, and the authors focus on an interesting research problem. I'd like to accept this paper and appreciate if the author can revise the paper more clearly.

---

> ### Author Response · Authors · 2022-11-14
> **Response to Reviewer rQ2x**
>
> We thank the reviewer for a positive review of our submission. The main concern is addressed below
>
> > This paper is not friendly to most beginners, and the main idea is hard to follow. The authors also need to explain more detailedly how the residual estimation can predict the performance.
>
> Thank you for the suggestion. In addition to the background in Section 2, we will add a more detailed background on harmonic analysis in the Appendix. The residual estimated by Algorithm 1 leverages the connection between the Fourier coefficients, the residual of linear datamodels in Theorem 2.2 Equation 4 and the noise stability in Equation 7. In particular, Theorem 2.2 shows that the residual is simply the mass of the Fourier coefficients of sets of size at least 2, leveraging the orthonormality of the Fourier basis. Since the mass of Fourier coefficients of sets of size $i$ shows up as the coefficients of $\rho^{i}$ in $h(\rho)$ (Equation 7), we can perform a polynomial approximation on $h$ to recover the required coefficients sums in order to estimate the residual. Our convex program in Algorithm 1 does precisely this (we just need to estimate the constant and linear coefficients to get the sum of the rest of the coefficients). Hope this provides more intuition for the residual estimation procedure.

---

> > ### Comment · Reviewer_rQ2x · 2022-12-02
> > **Reply to authors**
> >
> > Thanks for the clarification and I would like to maintain my positive score. I am looking forward to seeing the final version.

---

### Official Review · Reviewer_TeFi · 2022-10-23

**Confidence:** 2
**Correctness:** 4
**Technical Novelty And Significance:** 3
**Empirical Novelty And Significance:** 3
**Recommendation:** 6

**Clarity, Quality, Novelty And Reproducibility:**

The paper is well-written and clear. However, as mentioned in the previous section, it would be beneficial to give a more informative background about the theoretical tools used in the paper.

**Strength And Weaknesses:**

Strengths:

	- While the framework of datamodels empirically approximate training data influence, the original paper does not show any understanding of why it works in the first place. This paper, provides a starting point to analyse datamodels and further understand why it works. This is one of the major strengths of the paper.

	- The noise stability estimator is a good contribution to bypass the expensive process of training the datamodels (though see Weakness for a follow-up question).


While I am not an expert in harmonic analysis, there are certain questions regarding the paper:

Weakness / Questions:

	- While Ilyas et. Al (2022), show that datamodels can approximate group influence (via linearity), as shown in previous works [1], this linearity assumption is not always true, especially when a large number of points are deleted. In such cases, how does datamodels relate to higher-order variants of influence which seem to be a better approximation for group influence?

	- The paper has a good theoretical contribution, however I would like to see some more empirical analysis on how the noise stability estimator is a good proxy for predicting the fit. One way could be to run similar experiments on datasets beyond CIFAR-10 (e.g., Imagenet derivatives) to strengthen it's effectiveness.

The paper is difficult to read for someone who has limited background on harmonic analysis; Considering influence functions is used by a mix of empirical + theoretical researchers, it would be good to provide a small background on the topic in the Appendix or in the main paper (e.g., a brief background of the theoretical tools used in your analysis).


[1]. https://arxiv.org/abs/1911.00418





**Summary Of The Paper:**

The paper uses theoretical tools from harmonic analysis and noise stability to explain the effectiveness of data-models introduced by Ilyas et. al (2022). In addition, the same theoretical components are used to analyze group influence and identify conditions when first-order influence (linear in terms of training data influence) can be used to characterize group influence.

**Summary Of The Review:**

Overall,  I feel that the theoretical contribution is a good initial step towards understanding data-models and finding links between influence estimation and datamodels. While the authors connect the theory to empirical findings (e.g., prediction of the fit), this section can be strengthened with more experiments on datasets beyond CIFAR-10.

Overall, the paper can be a good addition to iclr as it provides some theoretical footing connecting datamodels to  influence functions.

---

> ### Author Response · Authors · 2022-11-14
> **Response to Reviewer TeFi**
>
> We thank the reviewer for the positive response. We address the concerns below:
>
> > Ilyas et al show that datamodels can approximate group influence via linearity, but as shown in previous work (Basu et al. 20) this linearity assumption is not always true, especially when a large number of points are deleted. In such cases, how does the datamodel relate to higher order variants of influence?
>
> We describe the main differences between datamodels and the continuous influence notions. Illyas et al. found *experimentally* that their datamodels can provide good approximations for group influence in deep learning settings. Our results show that linear datamodels *provably* estimate the leave-one-out (LOO) notion of influence (Proposition 2.1) and we address group influence in Section 4, explaining in what sense linearity is (or is not) valid for group influence. Theorem 4.1 shows how higher order Fourier coefficients show up in the goodness of linearity of group influence.
>
> Since the reviewer asks about non-linear variants of influence, we note that continuous influence notions from Koh et al. ‘19 and the second-order paper (Basu et al. 20) were also used to estimate group influence. However Basu et al. 21 (a different paper) found that these methods fail to capture group influence beyond linear models, i.e. do not apply in deep learning settings. (Note "linear" here means linear in model parameters which is different from linear datamodels). In fact they found that continuous influence even fails to capture single point LOO influences. Bae et al.22 study this in more detail and find that continuous influences capture a different notion of influence: effect of deleting a point and *fine-tuning* from the current model rather than retraining from scratch (LOO). Thus the continuous notions do not effectively capture LOO influences and so there is no a priori connection between linear datamodels (which are LOO) and such methods. Hope this clarifies the differences; we will include a discussion about this in the revision.
>
>
> > The paper has a good theoretical contribution, however, I would like to see some more empirical analysis on how the noise stability estimator is a good proxy for predicting the fit, by running similar experiments on datasets beyond CIFAR-10 (e.g. ImageNet and derivatives).
>
> We would like to note that Illyas et al. also ran experiments on CIFAR-10 and FMoW (similarly sized). This is because ImageNet scale experiments are prohibitively expensive to run, even with their efficient ffcv implementations, since these experiments require training hundreds of thousands of neural nets. Our results show that our estimation procedure (Algorithm 1) has a better upper bound for the number of neural nets to be trained ($1/\epsilon^3$) compared to learning datamodels ($N/\epsilon^2$). So for a desired precision of $\epsilon$, our procedure should be asymptotically much better (when $N \gg 1/\epsilon$), where $N$ is the size of the training set being used. Thus we expect the benefit to be larger for larger datasets. We are working on finding a reasonable small scale setting where our procedure is also empirically better than learning datamodels, and hope to include it in the revision if that succeeds.
>
> > The paper is difficult to read for someone who has limited background on harmonic analysis; Considering influence functions is used by a mix of empirical + theoretical researchers, it would be good to provide a small background on the topic.
>
> Thank you for the suggestion. In addition to the background in Section 2, we will add a more detailed background on harmonic analysis in the Appendix.
>
> References
>
> Koh et al. On the Accuracy of Influence Functions for Measuring Group Effects. 2019
>
> Basu et al. On Second-Order Group Influence Functions for Black-Box Predictions. 2020
>
> Basu et al. Influence Functions in Deep Learning Are Fragile. 2021
>
> Bae et al. If influence functions are the answer, then what is the question? 2022

---

> > ### Comment · Reviewer_TeFi · 2022-11-16
> > **Reply to Authors**
> >
> > I thank the authors for their detailed reply. I would like to maintain my score and inclined towards accepting the paper.

---

### Official Review · Reviewer_ECUW · 2022-10-24

**Confidence:** 3
**Correctness:** 3
**Technical Novelty And Significance:** 3
**Empirical Novelty And Significance:** Not applicable
**Recommendation:** 5

**Clarity, Quality, Novelty And Reproducibility:**

This paper is written clearly and well organized. There are a few flaws. For example, in page 2 line 2, author used symbol ‘ith’ but ‘i-th’ in other parts to represent the same thing.

**Strength And Weaknesses:**

Strength:
1. This paper introduces harmonic analysis into the discussion of influence functions and datamodels, which is a novel idea.
2. This paper gives a new algorithm to estimate the degree whether a linear datamodel is well-approximated without having to train the datamodel per se.

Weakness:
1. To my knowledge, all the analysis in this paper is based on a fixed test point, which means that we need to repeat the algorithm proposed in this paper with $m$ times when we need to estimate $m$ test points.
2. In Section 3.1, the author claimed that normalized noise stability should be high, and perhaps close to its maximum value of $1$ when the number of training samples grows, intuitively. Then Theorem 3.2. gives a bound on the best linear approximation in terms of the magnitude of the residual error because $1- \overline{h} (\rho)$ is close to $0$. But if I understand correctly, $\overline{h} (\rho)$ is close to $1$ when $\rho$ is close to $1$ intuitively. The bound given in Theorem 3.2 is meaningless at this point. In consequence, I think this intuition lacks theoretical or experimental evidence. I am willing to raise my score if the author can explain this question theoretically or experimentally.
3. It becomes worthless to estimate the quality of the best linear datamodel when it becomes a question whether datamodel is a good approximation.

**Summary Of The Paper:**

This paper provides some mathematical understanding on influence functions and datamodels via harmonic analysis including
1. reasons for existence of datamodels.
2. exact characterizations of \theta_i for datamodels without/with abbitrary regularization.
3. providing a new algorithm to estimate the degree whether a linear datamodel is well-approximated without having to train the datamodel per se.
4. studing group influence which quantifies the effect of adding or deleting a set.

**Summary Of The Review:**

This paper introduces harmonic analysis into the discussion of influence functions and datamodels and gain some novel results. But the main theorem proofs based on the intuition. Generally, I think harmonic analysis is a useful and relatively novel tool for machine learning.

---

> ### Author Response · Authors · 2022-11-14
> **Response to Reviewer ECUW**
>
> We thank the reviewer for the positive review. The concerns and potential misunderstandings are addressed below:
>
> > In Section 3.1, the author claimed that normalized noise stability should be high, and perhaps close to its maximum value of  1 when the number of training samples grows, intuitively. ... But if I understand correctly, $\bar{h}(\rho)$ is close to 1 when $\rho$ is close to 1 intuitively. The bound given in Theorem 3.2 is meaningless at this point. In consequence, I think this intuition lacks theoretical or experimental evidence. I am willing to raise my score if the author can explain this question theoretically or experimentally.
>
> Response: We believe there is a misunderstanding. Here $\rho$ is **not** the fraction of datapoints used; that fraction is denoted by $p$. Here $\rho$ denotes the correlation between two datasets used in the noise stability calculations; see Definition 3.1. You are correct that $\bar{h}(\rho) = 1$ at $\rho=1$ but to compute the bound we have the freedom to compute $\bar{h}(\rho) = h(\rho) / h(1)$ for any $\rho$ in the range $[0,1]$ (In fact our better estimation method of Section 3.2 leverages multiple $\rho$'s. ) The main insight in our calculation is that if $\bar{h}(\rho)$ is close to 1 for a *small* value of $\rho$ (i.e., when datasets $x, x'$ are not highly correlated) then we get a meaningful upper bound. We intuitively explain why $\bar{h}(\rho)$ ought to be small in practice for small $\rho$: when the number of training examples is large, even if two random datasets are not very correlated, the models trained using them should have similar predictions on test points. We hope this clarifies the relevance of the bound in Theorem 3.2.
>
> > All the analysis in this paper is based on a fixed test point, which means that we need to repeat the algorithm proposed in this paper m times when we need to estimate m test points.
>
> Response: No, our general analysis allows $f(x)$ to be any function of the net trained on the training set $x$. (e.g., it could be the average test error). The reviewer is referring perhaps to a specific application in Illyas et al. where $f(x)$ = error on a specific test point after training on $x$, and so they learn a different datamodel for each test point. (Note however that the computationally expensive part is training neural nets for many $x$. However this only needs to be done once. Training datamodels for any desired $f$ is then a Lasso regression that only requires inference with these trained nets.)
>
> Our paper characterizes rigorously what the datamodel parameters mean —they’re estimating leave-one-out influence.  One novel algorithmic contribution is Algorithm 1, which estimates the error of the linear data model via a convex program with 2-3 variables. For an $f$ that is defined for a single test point, running our convex program is significantly more efficient than learning a datamodel once we have the trained nets. Furthermore our method requires much fewer trained neural nets compared to Illyas et al., as quantified in Theorem 3.3. We will clarify these point in the revision.
>
>
> > It becomes worthless to estimate the quality of the best linear datamodel when it becomes a question whether datamodel is a good approximation.
>
> The following may address the reviewer's confusion.  Ilyas et al. Neurips'22 experimentally showed, for a couple of datasets, that linear data models are good approximations to the result of training on a dataset $x$. This is a surprising finding because a priori we do not expect the result of deep learning on a dataset $x$ to be expressible as a linear function of $x$. Our paper gives an intuitive yet rigorous explanation for this phenomenon for general datasets using harmonic analysis, as summarized in Section 1.1.

---

> ### Author Response · Authors · 2022-12-05
> **Any further questions/clarifications?**
>
> Thank you for the comments and questions. We hope that our response satisfactorily addressed all of your questions and concerns. Particularly with regards to "weakness 2", we hope that our explanation of Theorem 3.2 convinces you to raise your score, as you suggested in your review. We would be happy to provide more clarification if needed.

---

### Decision · Program_Chairs · 2023-01-20

**Decision:**

Accept: poster

**Justification For Why Not Higher Score:**

The paper focuses on harmonic analysis, a tool that might not be known to a wide audience at ICLR. Additionally, while the theoretical idea is interesting, the limitations of the empirical section (which are fully understandable given the computational cost) might limit its appeal.

**Justification For Why Not Lower Score:**

This is one more tool to provide a functional analysis of large models and attempts to predict their behaviour in increasingly common scenarios. Accepting this paper will provide a refreshing perspective on this question.

**Metareview: Summary, Strengths And Weaknesses:**

This paper proposes a harmonic analysis perspective on influence functions and datamodels. Even though the reviewers agreed that the empirical performance was not necessarily groundbreaking, they all appreciated this new perspective.

In an era where models are trained on increasingly large datasets and adding and removing examples are an important way to tune them with minimum additional computation, I welcome any attempt at providing more understanding of the influence of such changes on the final model.

**Note From Pc:**

if the above contains the word "oral" or "spotlight" please see: "oral" presentation means -> notable-top-5% and "spotlight" means -> notable-top-25%. As stated in our emails, we are disassociating presentation type from AC recommendations

**Summary Of Ac-Reviewer Meeting:**

N/A